



# Observational-based quantification of physical processes that impact the evolution of global mean sea level

Sjoerd Groeskamp[1]

[1]Royal NIOZ Netherlands Institute for Sea Research, 't Horntje (Texel), Netherlands

**Correspondence:** Sjoerd Groeskamp (sjoerd.groeskamp@nioz.nl)

**Abstract.** The global mean sea level (GMSL) rise budget can be closed by integrating over global density distribution to account for ocean expansion, combined with satellite altimetry and ice sheet mass balances obtained from satellites and models. Such methods are of great importance, but the disadvantage is that they gloss over the individual fundamental physical processes that impact the evolution of GMSL. Such process are for example (but not limited to) the impact of ocean mixing and stirring, neutral physics, shortwave radiation and boundary heat and freshwater fluxes. It is valuable to quantify these processes as it provides understanding in the fundamental processes behind the observed GMSL rise and how these processes may change and in a transient ocean and climate. This study estimates the contribution of individual physical processes contributing to GMSL rise, using observational based products. It is not the intention and neither possible at this stage, to close the GMSL budget by means of this approach. Instead the results allow us to gain insights in the magnitude and uncertainty of processes and their relative importance in shaping GMSL rise, and allows for a comparison of the impact on GMSL by single processes or parameterizations. Results indicate the great uncertainty related to boundary heat, mass and freshwater fluxes, and the importance of ocean mixing for GMSL rise. Unexpected results are the significant impact of shortwave radiation depth penetration parameterizations and the way by which neutral physics is implemented. Many of the results are of importance for observational-based calculations as well as for modelers that have specific choices to make about which method and parameterizations they choose. These choices significantly impact the accuracy of predicted future sea level rise upon which policy will be based.

## 1 Introduction

Since 1850 human induced anthropogenic warming has caused global surface temperatures to increase by about $1.1\,°C$ (IPCC, 2021). Yet, 89% of human induced warming of the climate system (anthropogenic heat) is absorbed by the ocean (von Schuckmann et al., 2020) leading to significant ocean warming (Li et al., 2023), thermal expansion and associated sea level rise (Horwath et al., 2022). About 4% of the warming since pre-industrial era has gone into melting glaciers and ice sheets, also elevating global sea level.

Since 1901 global mean sea level (GMSL) has risen by 20 cm (Frederikse et al., 2020). Sea level rise has accelerated from 1.35 mm mm year$^{-1}$ between 1901 and 1990 to 3.7 mm year$^{-1}$ between 2006 and 2018 (chapter 9, table 9.5, Masson-Delmotteet al. (2021)). Currently the main contributions to sea level rise are from thermal expansion (39% , 1.4 mm year$^{-1}$





) and the sum of glaciers and ice sheets (45%, 1.6 mm year$^{-1}$). Such sea level rise rates are obtained using a "top-down"

approach, by measuring GMSL changes by integrating over global temperature and salinity budgets (Pattullo et al., 1955;
Antonov et al., 2002), using satellite altimetry (Ablain et al., 2017) and ice sheet mass balances obtained from satellites and
models (Team, 2018). This approach has allowed to understand and close the GMSL budgets within the statistical uncertainty
range (Horwath et al., 2022; Frederikse et al., 2020; Church et al., 2011).

In addition to the above, I here apply a more "bottom-up" approach. That is, GMSL is calculated by summing up the underlying processes
that change ocean density and therewith ocean volume and GMSL. Such "non-Boussinesq" steric effects are quantified and
decomposed into contributions from boundary mass, heat and salt fluxes and that due to interior diffusive and advective fluxes
as a result of oceanic turbulent eddies. This approach was used by Griffies and Greatbatch (2012) within a numerical model
environment, with which they were able to close the model's GMSL budget. This study instead uses observational based
products. Note that it is not the intention, neither possible at this stage, to observationally close the GMSL by means of
this bottom-up approach. However, this approach does allow for a comparison of the impact on GMSL by single processes
or parameterizations. Such experiments are much harder to perform when using numerical global ocean model, due to the
complexity and expense of reprogramming and rerunning the model many times, and because it can be difficult in numerical
ocean models to isolate the impact of a single change.

In this work I will quantify the magnitude of a wide variety of processes influencing the GMSL, and gauge how uncertainties
in their observations or parameterizations influence the calculated GMSL budget. The focus is foremost on different air-sea flux
products, mixing strength parameterizations, neutral physics, shortwave radiation depth penetration parameterisations and eddy
stirring parameterizations. All processes mentioned above turn out to impact GMSL rise by leading order, that is sea level rise
of several mm year$^{-1}$. These results are of interest for numerical climate models that have to make choices on how to represent
all these processes, which raises question about how such choices influence calculations of future sea level rise. Result show
that it is not yet possible to close the GMSL budget using this bottom-up approach due to both a lack of observations as well
as representation of some of the fundamental underlying physical mechanisms.

In addition to the above, I will also explore if "densification upon mixing" is indeed able to keep the ocean from blowing up
(Schanze and Schmitt, 2013). That is, even without anthropogenic ocean warming, i.e. for an ocean with a net-zero globally
integrated heat flux, the ocean volume would still expand and sea level would still rise. This is because the thermal expansion
coefficient $\alpha$ is a nonlinear function of temperature (and to a lesser extend salinity and pressure). This results in $\alpha$ being
larger for warmer water than for colder water. As ocean warming mainly occurs at low latitudes over warm water (around the
equator), while cooling mainly occurs over higher latitudes over colder water (towards the poles), this leads to more expansion
by warming than there is contraction by cooling. The resulting net increase in ocean volume turns out to have a first-order
contribution to the GMSL budget (Griffies and Greatbatch, 2012; Schanze and Schmitt, 2013). I will here refer to this process
as "nonlinear thermal expansion" as it occurs only due to nonlinearity of the equation of state, and in addition to the thermal
expansion that is caused by having a globally integrated net heat flux into the ocean due to climate change. Opposing nonlinear
thermal expansion is densification upon mixing, that can lead to sea level decline (Gille, 2004; Jayne et al., 2004), emphasizing
the importance of mixing processes on sea level rise. Densification upon mixing occurs when two water parcels mix their




temperature and salinity, and the resulting mean density is denser than the average of the original densities due to the nonlinear equation of state. This process, also know as cabbeling (Witte, 1902; Foster, 1972; McDougall, 1987b), causes the water to contract, the volume to reduce and therefore sea level to fall. Hence, as first suggested by McDougall and Garrett (1992), densification upon mixing must oppose nonlinear thermal expansion and therewith keeps the ocean from ever expanding.

Although Schanze and Schmitt (2013) estimated the magnitude of densification upon mixing in the ocean based on air-sea heat fluxes, it has thus far not been verified by calculating the individual terms separately from observations, as done in this study.

This paper is structured as follows. In section 2 of this paper, the equations governing the physical oceanographic processes influencing the GMSL budget are presented and derived, also providing context on the assumption made and processes not covered in this study. Section 3 then describes the datasets used. In section 4 the results are shown for the nonlinear thermal

expansion (section 4.3), densification upon mixing (section 4.6), shortwave radiation (section 4.4) and stirring (section 4.7), amongs others. These results are presented for both the spatial distribution of all these processes as well as their net impact on GMSL. After a summary and discussion of the caveats and consequences (section 5) a brief conclusion is presented (section 6).

## 2 Theory

Sea level changes due to adding or redistributing mass, and due to changes in ocean density (Gill and Niller, 1973). Following Griffies and Greatbatch (2012) (their Eq. 1), these effects are described using the kinematic equation for sea level evolution:

$$\underbrace{\frac{\partial \eta}{\partial t}}_{\text{sea level evolution}} = \underbrace{\frac{Q_{\text{mass}}}{\rho(\eta)}}_{\text{boundary mass flux}} \underbrace{-\nabla \cdot \int_{-H}^{\eta} \mathbf{u}\, dz}_{\text{dynamic changes}} \underbrace{- \int_{-H}^{\eta} \frac{1}{\rho} \frac{d\rho}{dt}}_{\text{Non-Boussinesq steric}}. \tag{1}$$

Here the sea level evolution (l.h.s.) is given by the sum of boundary mass fluxes, redistribution of ocean volume by ocean currents and due to "Non-Boussinesq" steric changes, i.e. expansion or contraction of seawater due to changes in density. The

latter term contains the effects of for example air-sea fluxes, ocean mixing and geothermal heating, as detailed in Appendix A. Eq. 1 is derived under the assumptions that the ocean bottom does not move, the ocean surface area is constant and that the gravitational acceleration is constant (Griffies and Greatbatch, 2012). In addition, there are processes that are not covered here, such as the inverse barometric effect or the tidal sea surface elevations and joule heating (Griffies and Greatbatch, 2012).

In this study both the spatial structure of sea level evolution, as well as the net GMSL changes is studied. The spatial

structure is given by applying Eq. 1 to a vertical water column at each location, while its impact on GMSL rise $\frac{\partial \overline{\eta}}{\partial t}$ is obtained by integrating those results globally:

$$\frac{\partial \overline{\eta}}{\partial t} = \frac{1}{\mathcal{A}} \int_{\text{globe}} \frac{\partial \eta}{\partial t}\, dA, \quad \text{with} \quad \mathcal{A} = \int_{\text{globe}} dA. \tag{2}$$

Note that under a no-flux boundary condition, the term coined "dynamic changes" of Eq. 1 vanishes (second term, r.h.s.). This means that ocean currents have no net effect on GMSL rise and instead only redistribute volume.




In the remainder of this section the boundary mass fluxes (section 2.1) and redistribution of ocean volume by ocean currents (section 2.4) are specified and derived. However, the focus of this study is on "non-Boussinesq" steric changes due to diffusive salt and heat fluxes (section 2.3), skew fluxes of salt and heat (section(2.4), and by the combination of direct and indirect boundary fluxes of salt and heat (section 2.2). The quantification of all these terms are given in section 4. Derivations rely heavy on (Griffies and Greatbatch, 2012) and (Groeskamp et al., 2019b). To shorten the main text the details of most derivations are moved to the appendix. Section 2 mostly summarizes the main points or new points, required for understanding the results.

## 2.1 Sea level change due to a boundary mass flux

At the ocean boundaries, the ocean mass flux is defined as ($Q_\mathrm{mass}$, in kg s$^{-1}$ m$^{-2}$) positive into the ocean and given by:

$$Q_\mathrm{mass} = P - E + R + I + A_\mathrm{e}. \tag{3}$$

Where evaporation $E$ is positive out of the ocean, precipitation $P$ is positive into the ocean, runoff from ice melt $I$ or rivers $R$ are positive into the ocean and aeolian deposition of salt ($A_\mathrm{e}$) is positive into the ocean. These mass fluxes can enter the ocean by crossing the ocean surface ($E$, $P$, and $A_\mathrm{e}$), laterally ($R$) or from both directions ($I$). Inserting Eq. 3 into Eq. 1 leaves:

$$\left.\frac{\partial \eta}{\partial t}\right|_\mathrm{mass} = \frac{Q_\mathrm{mass}}{\rho(\eta)} = \frac{1}{\rho(\eta)}\left(P - E + R + I + A_\mathrm{e}\right). \tag{4}$$

Note that even for a net-zero global mean mass flux, there can still be a net contribution to sea level rise, as the impact of the mass flux on volume is weighted by the sea surface density $\rho(\eta)$ (section 4.1). This effect is conceptual similar to the impact of a changing thermal expansion coefficient for a net-zero global heat-flux as previously mentioned in section 1 and detailed in Appendix B

## 2.2 Sea level change due to boundary heat, salt and freshwater fluxes

Boundary mass fluxes into the ocean (e.g., evaporation, precipitation, ice melt), as well as direct sources of heat and salt, all impact density and sea level. The impact of such boundary fluxes on sea level rise can be expressed as:

$$\left.\frac{\partial \eta}{\partial t}\right|_\mathrm{boundary} = -\int_H^\eta \frac{1}{\rho}\left(F^\rho_\mathrm{mass} + F^\rho_\mathrm{surface} + F^\rho_\mathrm{geo} + F^\rho_\mathrm{swr}\right) dz \tag{5}$$

Exact details for deriving Eq. (5) can be found in Appendix A and B. Here $F^\rho_\mathrm{mass}$ (kg m$^{-3}$ s$^{-1}$) contains the impact that the ocean mass flux $Q_\mathrm{mass}$ has on changing local temperature and salinity and therewith density. Mainly by altering salinity. Here $F^\rho_\mathrm{surface}$ (kg m$^{-3}$ s$^{-1}$) captures the impact on density by direct sources of salinity and heat at the surface of the ocean. Such surface heat fluxes are longwave radiation as well as latent and sensible heat fluxes Shortwave radiation (SWR) enters the ocean surface and can penetrate to deeper layers depending on the clarity of the water (Paulson and Simpson, 1977). Its effect on density is represented by $F^\rho_\mathrm{swr}$ (kg m$^{-3}$ s$^{-1}$). Surface salt fluxes are associated with for example sea ice or spray and will be ignored in this study. Meanwhile $F^\rho_\mathrm{geo}$ captures geothermal heating at the sea floor that plays a small role in altering the density budget (de Lavergne et al., 2015).





## 2.3 Sea Level Change due to diffusive fluxes

In this section an expression is derived for the impact of diffusive mixing on density and sea level. Mixing is split into a contribution from mesoscale and small-scale processes (Fox-Kemper et al., 2019). Here small-scale mixing are due to eddies $\mathcal{O}$(meter) associated with for example breaking internal waves and boundary-layer processes (MacKinnon et al., 2013; Large

et al., 1994) and is simplified using a turbulent diffusivity $D$ acting on vertical tracer gradients (McDougall et al., 2014). The magnitude of the vertical eddy diffusivity is typically $\mathcal{O}$ ($10^{-5}$ - $10^{-3}$ m$^2$ s$^{-1}$ ) (Polzin et al., 1997; Whalen et al., 2012; de Lavergne et al., 2020). Mesoscale eddies $\mathcal{O}$(20-200 km) that stir tracers along neutral directions are parameterized by isoneutral eddy diffusivity $K_N$ acting on tracer gradient along a neutral direction $\nabla_N C$ (Redi, 1982; Griffies, 1998; McDougall et al., 2014). When influenced by the geometric constraints of the surface boundary, mesoscale stirring leads to horizontally

oriented mixing across outcropped density surfaces (Tandon and Garrett, 1997; Treguier et al., 1997; Ferrari et al., 2008), which is parameterized by a horizontal diffusivity $K_H$ acting on horizontal tracer gradient $\nabla_H C$. The magnitude for $K_N$ and $K_H$ is typically $\mathcal{O}$ ($10^1$ - $10^3$ m$^2$ s$^{-1}$ ) (Abernathey and Marshall, 2013; Klocker and Abernathey, 2013; Cole et al., 2015; Roach et al., 2018; Groeskamp et al., 2017, 2020).

The above directional and scale variations for mixing will here be represented in mixing tensor $\mathbf{K}$ (m$^2$ s$^{-1}$) as a symmetric

positive-definite kinematic diffusivity tensor that contains the contributions of the mesoscale neutral and horizontal diffusion, and small-scale isotropic diffusion. This leads to the following expression for the impact of diffusive mixing on density and sea level, for which the details are provided in Appendix C:

$$
\left.\frac{\partial \eta}{\partial t}\right|_{\text{diffusion}} \approx \underbrace{-\nabla_H \cdot \int_{-H}^{\eta} \mathbf{R}^{\text{hor}} \, dz}_{\text{Redistribution}} + \underbrace{\mathbf{R}^{\text{hor}}(\eta) \cdot \nabla_H \eta - \mathbf{R}^{\text{hor}}(-H) \cdot \nabla_H(-H)}_{\text{diffusion−boundary interaction}}
$$

$$
+ \underbrace{\int_{-H}^{\eta} K_H^{-1} \left|\mathbf{R}^{\text{hor}}\right|^2 \, dz + \int_{-H}^{\eta} D^{-1} \left|\mathbf{R}^{\text{ver}}\right|^2 \, dz - \int_{-H}^{\eta} \rho g \kappa \mathbf{k} \cdot \mathbf{R}^{\text{ver}} \, dz}_{\text{diffusion−density interaction}}
$$

$$
+ \underbrace{\int_{-H}^{\eta} P_{T_b}^{\text{ntr}} + P_{C_b}^{\text{ntr}} + P_{C_b}^{\text{hor}} + P_{T_b}^{\text{ver}} + P_{C_b}^{\text{ver}} dz}_{\text{Cabbeling and Thermobaricity}}. \tag{6}
$$

Where used the following definitions (see Appendix C)

$$
\mathbf{R} = -\alpha \mathbf{K} \cdot \nabla\Theta + \beta \mathbf{K} \cdot \nabla S_A \tag{7}
$$

$$
P = \nabla\alpha \cdot (\mathbf{K} \cdot \nabla\Theta) - \nabla\beta \cdot (\mathbf{K} \cdot \nabla S_A) \tag{8}
$$

Here $\mathbf{R}^{\text{ntr}}$, $\mathbf{R}^{\text{hor}}$, and $\mathbf{R}^{\text{ver}}$ (m s$^{-1}$) are the components of $\mathbf{R}$ for the three different mixing direction, while $P^{\text{ntr}}$, $P^{\text{hor}}$, and $P^{\text{ver}}$ are the components of P (s$^{-1}$) for the three different mixing direction. For the neutral direction $\alpha \nabla_N \Theta = \beta \nabla_N S_A$, and thus $\mathbf{R}^{\text{ntr}} = 0$. The first term on the r.h.s. of Eq. 6 named "redistribution" turns out to be large, but also globally integrates to zero




due to the divergence operator. By explicitly writing Eq. 6 in a form that contains this "redistribution" term also creates the term
named "diffusion-density interaction" that can be interpret as interaction between diffusion and density gradients. This term
will turn out to be small, such that it is advantages to write Eq. 6 in this form, as it allows for neglecting the global integration
of redistribution term and the diffusion-density interaction term. The term named "Production" is the interaction between
diffusion and the nonlinear equation of state. This term includes the terms causing densification upon mixing, cabbeling and

the thermobaric terms, as specified in Appendix C1. Thermobaricity can in fact lead to both an increase and decrease in density,
but its impact is on density is generally an order of magnitude smaller than that due to cabbeling (Klocker and McDougall, 2010;
Groeskamp et al., 2016). Although cabbeling and thermobaricity are names specifically referring to neutral mixing (McDougall,
1987b), effects of the nonlinear equation of state that change density due to mixing of $\Theta$ and $S_A$, are not limited to the neutral
direction. Here the same naming is used for the impact of cabbeling ($P_{C_b}^{ntr}, P_{C_b}^{hor}, P_{C_b}^{ver}$) and thermobaricity ($P_{T_b}^{ntr}, P_{T_b}^{hor}, P_{T_b}^{ver}$) for

all three mixing direction.

### 2.4    Sea level change due to ocean dynamics and eddy-induced transport

The convergence of the vertically integrated horizontal ocean velocity $\mathbf{u}$ can lead to redistribution of volume and thus a local
impact on sea level, while its global integral is zero under the assumption that there is no velocity through the boundaries. By
approximating the velocity by the sum of the geostrophic flow and the eddy-induced velocity one obtains:

$$\mathbf{u} = \mathbf{u}_{geo} + \mathbf{u}_{eddy}, \tag{9}$$

where $\mathbf{u}_{geo}$ is the geostrophic velocity and $\mathbf{u}_{eddy}$ is the eddy-induced velocity. Following Ferrari et al. (2010), the eddy velocity
paramterization is constructed to ensure a net zero vertical integral over the eddy velocity in Eq. 1 (Appendix D), leaving:

$$\left.\frac{\partial \eta}{\partial t}\right|_{dynamic} = -\nabla \cdot \int_{-H}^{z} \mathbf{u}_{geo} \, dz. \tag{10}$$

Hence the impact of dynamic changes on the GMSL budget can locally be estimated using the thermal wind balance in

combination with a reference level velocity and will be zero when globally integrated.

Although the eddy-induced transport itself does not change sea level, it does impact density through non-resolved trans-
portation of salt and heat (Griffies and Greatbatch, 2012). The resulting impact of stirring on sea level evolution is given
by:

$$\left.\frac{\partial \eta}{\partial t}\right|_{stirring} = \underbrace{-\nabla_H \cdot \int_{-H}^{\eta} \mathbf{R}_{stir} \, dz}_{Redistribution} + \underbrace{\int_{-H}^{\eta} P_{stir} \, dz}_{Production} + \underbrace{\int_{-H}^{\eta} \mathbf{R}_{stir} \cdot \nabla \ln \rho \, dz}_{Eddy-density\ interaction}. \tag{11}$$

The full derivation for obtaining Eq. 11 is given in Appendix D. Note that the down gradient eddy tracer flux of temperature
and salinity are embedded inside $\mathbf{R}_{stir}$ (m $s^{-1}$) and $P_{stir}$ ($s^{-1}$) in a manner comparable to that for diffusion:

$$\mathbf{R}_{stir} = -\alpha \mathbf{K}_{stir} \cdot \nabla \Theta + \beta \mathbf{K}_{stir} \cdot \nabla S_A, \tag{12}$$

$$P_{stir} = -\nabla \alpha \cdot (\mathbf{K}_{stir} \cdot \nabla \Theta) + \nabla \beta \cdot (\mathbf{K}_{stir} \cdot \nabla S_A). \tag{13}$$





Here $\mathbf{K}_{\mathrm{stir}}$ is the stirring strength operator (Eq. D3), also know as the "GM diffusivity" (Gent and McWilliams, 1990). The first term on the r.h.s. of Eq. 11 named "redistribution" globally integrates to zero, while the last term, named "Eddy-density interaction" will be small. Hence, it is the term named "Production" in Eq. 11 that will lead to the main impact of stirring on
GMSL rise. This term is related to the interaction between stirring and the nonlinearity equation of state, comparable to the cabbeling and thermobaricity terms for diffusion.

## 2.5  Sea level change due to the Non-Neutrality term

A term that does not exist in the real ocean, but does exist in any calculation involving neutral mixing, is the the "non-neutrality" term related to neutral physics (Griffies, 2004; Klocker et al., 2009). Diffusive fluxes and stirring in the neutral direction are
calculated using the slopes of the neutral tangent plane, and the tracer gradients along the neutral tangent plane (McDougall, 1987a). However, a neutral tangent plane is only locally defined (McDougall and Jackett, 1988), such that any calculated neutral slope or gradient will have an implicit error that depends strongly on the method used (McDougall and Jackett, 1988; Stanley, 2019; Stanley et al., 2020; Groeskamp et al., 2019a). The non-exact neutral gradients lead to fictitious diffusion (Klocker et al., 2009) that impacts density and sea level and could therefore cause additional densification upon mixing (Groeskamp et al.,
2019a). Acknowledging this error, leaves that $\mathbf{R}^{\mathrm{ntr}} \neq 0$, that the $\mathrm{P}^{\mathrm{ntr}}$ terms also have error embedded, and that the neutral slopes in the stirring term are not exact. The non-neutrality term is defines as:

$$
\begin{aligned}
\left.\frac{\partial \eta}{\partial t}\right|_{\mathrm{non-neutral}} \approx \quad & \underbrace{-\nabla_{\mathrm{H}} \cdot \int_{-H}^{\eta} \mathbf{R}^{\mathrm{ntr}} \, dz - \mathbf{R}^{\mathrm{ntr}}(-H) \cdot \nabla_{\mathrm{H}}(-H)}_{\mathrm{Redistribution}} \\
& + \int_{-H}^{\eta} K_{\mathrm{N}}^{-1} \left|\mathbf{R}^{\mathrm{ntr}}\right|^2 \, dz + \int_{-H}^{\eta} \kappa \nabla_{\mathrm{N}} P \cdot \mathbf{R}^{\mathrm{ntr}} \, dz \\
& + \int_{-H}^{\eta} \mathrm{P}^{\mathrm{ntr}} - \mathrm{P}^{(\mathrm{perfectly\ neutral})} \, dz + \left.\frac{\partial \eta}{\partial t}\right|_{\mathrm{stirring}}^{\mathrm{non-neutral}} .
\end{aligned}
\tag{14}
$$

Here $\mathbf{R}^{\mathrm{ntr}}(\eta) \cdot \nabla_{\mathrm{H}} \eta = 0$ as there are no neutral slopes at the surface, and $\mathrm{P}^{(\mathrm{ntr})} = \mathrm{P}^{(\mathrm{perfectly\ neutral})} + \mathrm{P}^{(\mathrm{error})}$. Hence, $\mathrm{P}^{(\mathrm{error})}$ is the impact of incorrect neutral physics calculation scheme. If $\mathrm{P}^{(\mathrm{error})} > 0$ then the method underestimates neutral gradients, while for $\mathrm{P}^{(\mathrm{error})} < 0$ the method overestimates neutral gradients. In the latter case, which is most common, this can be interpret as enhanced vertical mixing and leads to extra non-realistic densification upon mixing. Note that the vertical component of the $\mathbf{R}^{\mathrm{ntr}} \cdot \mathbf{k}$ has no role to play at all, because there are no vertical gradients of either the bottom slope or the surface slope, and
this term is zero after vertical integration. The last term $\left.\frac{\partial \eta}{\partial t}\right|_{\mathrm{stirring}}^{\mathrm{non-neutral}}$ is the impact of stirring along non-neutral slopes, as detailed in Appendix D. For this term, an overestimate of the neutral slopes will also lead to more reduction in GMSL.





## 2.6 The total sea level rise equation

Collecting all terms, the evolution of sea level rise can be expressed as:

$$
\frac{\partial \eta}{\partial t} = \underbrace{\frac{\partial \eta}{\partial t}\bigg|_{\text{mass}}}_{Eq.4} + \underbrace{\frac{\partial \eta}{\partial t}\bigg|_{\text{boundary}}}_{Eq.5} + \underbrace{\frac{\partial \eta}{\partial t}\bigg|_{\text{diffusion}}}_{Eq.6} + \underbrace{\frac{\partial \eta}{\partial t}\bigg|_{\text{dynamic}}}_{Eq.10} + \underbrace{\frac{\partial \eta}{\partial t}\bigg|_{\text{stirring}}}_{Eq.11} + \underbrace{\frac{\partial \eta}{\partial t}\bigg|_{\text{non}-\text{neutral}}}_{Eq.14}
\tag{15}
$$

Each of these terms consist of many processes, emphasizing the number of fundamental processes that need to be understood in order to privde an accurate bottom-up calculation of the GMSL budget. It also showcases that sea level is constantly evolving both locally, as well as globally integrated. In section 3 the data used to calculate all these terms are described and discuss in section 4.

## 3 Data used

Here a range of observational based products are described, that are needed for calculating the terms in the GMSL budget as defined in section 2.

### 3.1 Gridded climatology

For the observational based climatology, World Ocean Atlas 2018 (Garcia et al., 2019) is used. This is a set of objectively analyzed ($1°$ grid) climatological fields of *in situ* temperature $t$, practical salinity $S_{\mathrm{p}}$, and other tracers at standard depth levels
for annual, seasonal, and monthly compositing periods for the world ocean sometimes referred to as a "standard year". Monthly means for the upper 1500m are used, while it is assumed that the deep ocean has little seasonal variation, such that seasonal means (repeated per quarter) are used for the interior (below 1500 m). Topographic gradients (e.g. $\nabla_{\mathrm{H}}(-H)$ in Eq. 6) are calculated using vertical derivatives from WOA column depths. TEOS-10 software (IOC et al., 2010; McDougall and Barker, 2011) is applied to convert the data to Absolute Salinity $S_{\mathrm{A}}$ and Conservative Temperature $\Theta$, pressure $P$ and to calculate
the mixed layer depth (de Boyer Montégut et al., 2004). Static stability ($N^2 > 0$ everywhere) is obtained using a minimal adjustment of $S_{\mathrm{A}}$ and $\Theta$ within the measurement error (Jackett and McDougall, 1995; Barker and McDougall, 2020). TEOS-10 software is used to calculate the expansion coefficients and their gradients, and the cabbeling and thermobaricity coefficients ($C_{\mathrm{b}}, T_{\mathrm{b}}$).

### 3.2 Diffusivities for diffusion and stirring

The mesoscale neutral $K_{\mathrm{N}}$ and horizontal $K_{\mathrm{H}}$ diffusivities are based on the product from Groeskamp et al. (2020). They provide global 3D observational based estimates of oceanic mesoscale diffusivity on a gridded climatology of WOA18 using a combination mixing length theory (Prandtl, 1925), mean flow suppression theory (Ferrari and Nikurashin, 2010; Klocker et al., 2012), and the theory of vertical modes (LaCasce and Groeskamp, 2020). As the diffusivity obtained by Groeskamp et al. (2020) is static, they are repeated for each month to obtain estimates for $K_{\mathrm{N}}$, $K_{\mathrm{H}}$ and $K_{\mathrm{stir}}$. To separate between neutral and



horizontal mesoscale mixing, a a step-wise change at the mixed layer depth is applied. Above the mixed layer depth mixing is represented by horizontal mesoscale mixing and below the mixed layer depth it is represented by neutral mixing. The same product is used to approximate the mesoscale stirring diffusivities $K_{\text{stir}}$, thus assuming that stirring diffusivities are equal to tracer diffusivities, even though it is known they are not everywhere the same (Smith and Marshall, 2009; Abernathey et al., 2013).

For vertical mixing diffusivities the product of de Lavergne et al. (2020) is used, which is a parameterization based on the turbulence production due to internal tides (de Lavergne et al., 2019, 2020). Note that this parameterisation does not include surface boundary layer mixing processes often parameterized with for example the K-profile parameterization (KPP) scheme (Large et al., 1994), which has a significant impact on the GMSL budget (Griffies and Greatbatch, 2012).

For comparison, I also make use of constant diffusivities of $D = 5 \times 10^{-5}$ m$^2$ s$^{-1}$, $K_{\text{N}} = K_{\text{stir}} = 300$ m$^2$ s$^{-1}$, $K_{\text{H}} = 750$
m$^2$ s$^{-1}$, while keeping the mixed layer depth as separation between horizontal and neutral mixing.

### 3.3 Neutral Slopes and gradients

For the calculation of neutral slopes and gradients two methods are applied. Traditionally neutral slopes and gradients are calculated using the "local" method that computes the ratio of the horizontal to vertical derivative of $\sigma_l$, i.e. the locally referenced potential density (Redi, 1982; Griffies, 1998). The resulting slopes are then combined with the local spatial gradients
of tracer, to calculate the neutral tracer gradients. This "local method" is problematic in regions of weak vertical stratification, consequently requiring a variety of ad-hoc regularization methods that can lead to rather nonphysical dependencies for the resulting neutral tracer gradients. To avoid such dependencies Groeskamp et al. (2019a) developed the "vertical nonlocal method" (VENM), which is a search algorithm that requires no ad-hoc regularization and significantly improves the numerical accuracy of estimates of neutral slopes and gradients, making it one of the most accurate methods available for calculating
neutral slopes and gradients from ocean observations. A comparable version of such a method has been implemented in the Modular Ocean Model Version 6 (MOM6, (Adcroft et al., 2019)) by Shao et al. (2020) However, most models still use the "local" method. All our main results are based on the VENM method. To calculate the non-neutrality term, the results from the VENM algorithm are considered "perfectly neutral" in Eq. 14, while using the results from the "local" method as "ntr". Howver, VENM is not perfectly neutral, such that the magnitude of the non-neutrality term calculated this way is interpret as
an order of magnitude estimate of how much impact different methods of calculating neutral slopes and gradients can have on GMSL rise.

### 3.4 Velocity and surface height gradients

To calculate ocean geostrophic velocity the GSW-Software 'gsw_geo_strf_dyn_height' is used to calculate the dynamic height anomaly streamfunction, and taken as input for gsw_geostrophic_velocity to calculate the resulting geostrophic velocities.
The 1000 dbar is used as reference level, with the associated reference level velocity taken from the YoMaHa'07 Argo float trajectories based estimates at 1000 db from (Lebedev et al., 2007).





For obtaining an estimate of sea surface height variation ($\eta$, as in Eq. 6) and associated gradients, 10 years of data is used from 2014 to 2023 of altimeter satellite based Global Ocean Gridded L4 Sea Surface Heights data provided by E.U. Copernicus Marine Service Information (CMEMS) (Services, https://doi.org/10.48670/moi-00148 (Accessed on 09-Dec-2024). This data

was then interpolated to the WOA grid, before calculating the gradients.

### 3.5    Boundary mass and heat fluxes

To represent surface mass fluxes and air-sea-ice interaction I chose to use two products described below.

The first product is the Objectively Analyzed air-sea Heat Fluxes (OA; Yu et al. (2008); Yu and Weller (2007)). The OA flux is constructed from optimal blending of satellite retrievals and three atmospheric reanalysis in combination with bulk

formula. OA is combined with surface radiation data from the International Satellite Cloud Climatology Project (Zhang et al., 1995, 2004, 2007) to provide the heat fluxes and evaporation from 1983–2006 Precipitation data accompanying the OA are obtained as the long-term (1981–2010) monthly means (2.5 degree grid) from the Global Precipitation Climatology Project Version 2.3 (Adler et al., 2003) and interpolated to the WOA grid. Runoff is obtained from time series (1900–2014) of monthly river flow from stations of the world's largest 925 rivers, which excludes contributions from Greenland and Antarctica (Dai,

2016). Long-term monthly means are calculated, and 50% of the outflow to the ocean is allocated at the river mouth, spreading the other 50% over the surrounding ocean. The runoff data set does not take into account unmeasured continental runoff and underground seepage, which could be of the same order of magnitude as the river runoff (CORE2 Global Air-Sea Flux Dataset) but spread over all global basins (Large and Yeager, 2004).

The second product is version 2 of Common Ocean Reference Experiment (CORE)-based product (Large and Yeager, 2009;

Yeager and Large, 2008; Danabasoglu et al., 2014). Here bulk formula are applied in combination with adjusted wind speed and humidity to decrease a global net imbalance from 30 W m$^{-1}$ to 2 W m$^{-1}$. CORE combines the Global Precipitation Climatology Project with other products to obtain their P values. Monthly mean values (1949–2006) are constructed for Latent Heat, Sensible Heat, Longwave radiation and shortwave radiation, evaporation and precipitation. Then a standard year is constructed by averaging for each calendar month for the WOA grid. Runoff is based on (Dai, 2016), but with an extra runoff

term that is added for Antarctica.

I chose OA flux and COREv2 because they have the largest (OA) and smallest (CORE) globally integrated net heat flux imbalance (Valdivieso et al., 2017) (their Fig. 2), making them suitable to quantify the impact that different heat flux products have on GMSL budgets. The two products do use overlapping datasets to determine global mass fluxes. However, as the impact of mass fluxes are of secondary interest in this study, I have instead chosen these products for their range in heat fluxes. In the

next subsection it will discuss how these products are adapted to be globally balanced.

For the geothermal heat flux product we use the product as described in de Lavergne et al. (2015), based on Goutorbe et al. (2011).





### 3.6 Balanced boundary mass and heat fluxes

To understand the order of magnitude of the nonlinear thermal expansion, this study uses an artificially balanced mass and heat
flux product that is applied to the 'standard year' climatology. This means that there is no net global mass or heat flux that
leads to, for example, climate change induces sea level rise. The exact procedure is given in Appendix E, but the overarching
idea is that the total global imbalance is redistributed over each grid point for all contributing fluxes, and proportional to the
magnitude of the local flux. This assumes that if at a given time and location the flux is large, that the error is also larger and
it can compensate a larger proportion of the total imbalance. The result is a global net-zero mass and heat flux. This method is
applied to both mass and heat fluxes of OA and CORE and referred to as the "balanced" products.

### 3.7 Shortwave radiation (SWR) depth penetration parameterization

It is important to properly model the vertical distribution of the SWR, as it may significantly alter density changes over a range
of depths instead of only at the surface (Iudicone et al., 2011; Groeskamp and Iudicone, 2018). This study compares the impact
of three different SWR depth penetration parameterizations. A function $F(z)$ (see Eq. B4 is used to redistribute the incoming
SWR with depth according to the following functions:

$$F_{\mathrm{SF}} = \begin{cases} 1 & \text{if} \quad z = 0 \\ 0 & \text{if} \quad z > 0 \end{cases} \tag{16}$$

$$F_{\mathrm{PS77}} = Re^{-\frac{z}{h_1}} + (1-R)e^{-\frac{z}{h_2}}, \tag{17}$$

$$F_{\mathrm{MA94}} = I_{\mathrm{IR}} + I_{\mathrm{VIS}} \left( c_1 e^{-\frac{z}{\zeta_1}} + c_2 e^{-\frac{z}{\zeta_2}} \right), \tag{18}$$

For $F_{\mathrm{SF}}$ all SWR is absorbed at the surface. $F_{\mathrm{PS77}}$ is an exponential decay function (Paulson and Simpson, 1977) in which $R =$
$0.58$, $h_1 = 0.35$ m, $h_2 = 23$ m, corresponding to Type-1 water from Jerlov (1968). For $F_{\mathrm{MA94}}$, $I_{\mathrm{IR}}$ and $I_{\mathrm{VIS}}$ are the infrared
and visible light components of the SWR, respectively (Morel and Antoine, 1994). All infrared radiation will be absorbed
within 2 m (within the upper bin of the data used in this study), $I_{\mathrm{IR}} = 0.43$ (Sweeney et al., 2005), where $I_{\mathrm{IR}} + I_{\mathrm{VIS}} = 1$. The
dependence of the depth penetration on Chlorophyll-a (Chl-a) for the visible component of the SWR, is included in the factors
$c_1$, $c_2$, $\zeta_1$ and $\zeta_2$ in the exponents (which can be found in Table 2 of Morel and Antoine (1994)), such that $c_1 + c_2 = 1$, while $\zeta_1$
and $\zeta_2$ are e-folding depths (like $h_1$ and $h_2$). The general results that are presented in this study and in particular in Table 1 use
the paramterization $F_{\mathrm{MA94}}$ from Morel and Antoine (1994) (Eq. 18). Differences with other parameterizations are discussed in
section 4.4. A Chl-a climatology is constructed using the same product as in (Groeskamp and Iudicone, 2018), based on a 9-km
resolution monthly mean Sea-viewing Wide Field-of-view Sensor data for the period 1997–2010 (Hu et al., 2012), which was
spatially averaged to the WOA climatology to calculate standard year of monthly means by time averaging for each calendar
month. This provides a good first order estimate of the decay factors in $F_{\mathrm{MA94}}$ for the purposes of this paper.



## 4 Results

This section discusses the results of the quantification of GMSL changes due to the different processes described in section 2,
in combination with the data described in section 3.

### 4.1 GMSL rise due to mass fluxes

Here the direct impact of ocean mass fluxes on sea level rise is discussed (section 2.1, Eq. 4). The indirect impact of mass
fluxes on the salinity budget (Eq. 5) are discussed in section 4.2. The largest contributions to GMSL is due to Precipitation $P$
(just above -1100 mm year$^{-1}$) and evaporation $E$ (just above 1100 mm year$^{-1}$), and are of opposite sign. Some precipitation
falls on land and enters back via river runoff, which is why precipitation is a bit smaller and the difference is about that from
river runoff of the order of 65-85 mm year$^{-1}$ (Table 1).

Resulting patterns of sea level evolution are a direct reflection of well-known evaporation, precipitation and runoff patterns
(Fig. 1). Sea level decreases in subtropical regions where evaporation dominates, while sea level increases at higher latitudes
and equatorial regions where precipitation dominates (Fig. 1d).

The difference in GMSL rise between CORE and OA is about 40 mm year$^{-1}$, 10 times the observed rates of current sea
level rise. Such difference can easily occur from the use of different bulk transfer algorithms, calibration protocols and reason
for which the product is designed. A part of the difference is likely because CORE has some added contribution of Antarctica
runoff that is not present in OA flux. In general ice sheet melt from Greenland and Antarctica are poorly represented in CORE
and OA, while they have a large contribution to the observed GMSL rise (Horwath et al., 2022). Both mass fluxes used in
this study do not specifically include the impact of terrestrial water storage variability (Hamlington et al., 2017), nor do they
include Aeolian mass fluxes into the ocean.

When examining the GMSL changes due to the constructed balanced mass flux products of CORE and OA (section 3.6 and
Appendix E), the net change in GMSL is of the order of 0.1 mm year$^{-1}$, similar to that found by (Griffies and Greatbatch,
2012) in a numerical model. This is a consequence of mass entering the ocean at higher densities (higher latitudes), while
leaving the ocean at lower densities (lower latitudes, Eq. 1).

### 4.2 GMSL rise due to the surface freshwater flux

The mass flux (Eq. 3) is also a freshwater flux (Eq. B1) that can be calculated into an equivalent salt-flux that alters salinity
and therewith density and sea level (section 2.2, Huang (1993); Nurser and Griffies (2019); Groeskamp et al. (2019b)). The
difference between the mass flux and salt flux is only a factor $\beta S_{\mathrm{A}}$, meaning that the impact of the salt fluxes on sea level is
almost 40 times smaller than the direct impact of the individual mass flux terms (Table 1). The impact on GMSL rise due to
freshening resulting from precipitation is about 27 mm year$^{-1}$, a bit smaller than the impact from evaporation of about -30 mm
year$^{-1}$ (Table 1). The residual is covered by the impact on freshening by river runoff (about 2 mm year$^{-1}$). Resulting patterns
of sea level evolution are different from $E$, $P$ and $R$ by the factor $\beta S_{\mathrm{A}}$, but overall comparable (Fig. 2) Hence, as for the mass



**Figure 1.** The spatial impact of mass fluxes on sea level rise (m s$^{-1}$) due to Evaporation (a), Precipitation (b), Runoff (c) and their sum (d). Derived and discussed in sections 2.1, 3.5, 4.1, Eq. 4 and Table 1. Runoff from ice melt $I$ or aeolian deposition of salt $A_e$ are neglected.





| GMSL in mm year$^{-1}$ | Normal | | Balanced | |
|---|---|---|---|---|
| | Core | OA | Core | OA |
| **Total Mass** | **-35** | **3.3** | **0.11** | **0.05** |
| Evaporation | -1185 | -1120 | -1167 | -1121 |
| Precipitation | 1064 | 1057 | 1080 | 1055 |
| Runoff | 85 | 66 | 87 | 66 |
| **Total Freshwater** | **-1.2** | **-0.24** | **-0.27** | **-0.33** |
| Evaporation | -30 | -29 | -30 | -29 |
| Precipitation | 27 | 27 | 28 | 27 |
| Runoff | 2.1 | 1.5 | 2.1 | 1.5 |
| **Total Heat** | **13** | **57** | **5.8** | **1.0** |
| Longwave | -99 | -92 | -101 | -100 |
| Shortwave | 339 | 356 | 335 | 326 |
| Latent heat | -203 | -189 | -205 | -205 |
| Sensible heat | -23 | -18 | -24 | -19 |

**Table 1.** Area weighted GMSL rise in mm year$^{-1}$, calculated using Eq. 2 for surface mass fluxes (first 4 rows), freshwater fluxes (row 5-8) and heat fluxes (row 10-14). Mass and freshwater fluxes are due to evaporation, precipitation and runoff, while the heat fluxes are due to longwave and shortwave radiative fluxes and sensible and latent heat turbulent fluxes. Bold indicate their sums. See also sections 2.2, 4.1-4.3, Eq. 4, 5 and Figs. 1, 2 and 3.

flux, sea level decreases in subtropical regions where evaporation dominates, while sea level increases at higher latitudes and
equatorial regions where precipitation dominates (Fig. 2d).

The net impact of freshwater fluxes are about -1 mm year$^{-1}$, with the difference between OA and CORE being about the same size and thus of the same order as observed GMSL rise. As for the balanced mass flux products, the resulting "nonlinear haline expansion" leads to GMSL change of about $-0.3$ mm year$^{-1}$ for both products. This is nonzero as the balanced mass flux is weighted by the factor $\beta S_{\mathrm{A}}$, which leads to a significant net GMSL rise, but still an order of magnitude smaller than
observed GMSL rise.

### 4.3 GMSL rise due to the surface heat flux

Here the effect of the sensible heat flux, latent heat flux and long-wave heat flux that are exchanged at the oceans surface (section 2.2), and the shortwave radiative heat flux that penetrates deeper into the ocean (Table 1 and Fig. 3), are calculated. The shortwave heat flux leads to an increase in sea level, where the other heat fluxes lead to a decrease. Their individual
impacts on GMSL are among the highest of all terms and vary between -200 mm year$^{-1}$ to 350 mm year$^{-1}$(Table 1). The total net impact of heat flux is about 10-50 mm year$^{-1}$, i.e. with a difference between the two products of about 40 mm year$^{-1}$, comparable to that due to global mass fluxes. This difference is not unexpected as heat flux products are notoriously difficult

**Figure 2.** The spatial impact of freshwater fluxes on sea level rise (m s$^{-1}$) due to Evaporation (a), Precipitation (b), Runoff (c) and their sum (d). Derived and discussed in sections 2.2, 3.5, 4.2, Eq. 5 and Table 1. Runoff from ice melt $I$ or aeolian deposition of salt $A_\mathrm{e}$ are neglected.





to close (Josey et al., 1999; Yu, 2019). A net global heat flux of 0.3 W m$^{-2}$ is enough to explain the observed increase in global heat content (Domingues et al., 2008; Ishii and Kimoto, 2009; Levitus et al., 2012; Lyman and Johnson, 2013), while

imbalances a 100 times larger are not uncommon for available heat flux products (Balmaseda et al., 2015; Valdivieso et al., 2017). The CORE and OA flux products have some of the largest difference in net heat flux (Valdivieso et al., 2017), meaning that these results should be a good indication of the range that can be expected from the impact of heat fluxes on GMSL.

Resulting patterns of sea level evolution (Fig. 3) are a direct reflection of heat flux patterns themselves that can be found in many different studies (Josey et al., 1999; Yu, 2019; Tang et al., 2024). Of interest is the distribution of these fluxes, showing

warming around the equator and cooling at western boundary currents and higher latitudes (Fig. 3e). As the thermal expansion coefficient can vary up to a factor 10 (especially with latitude), this leads to differently weighted impact of warming and cooling on sea level.

Nonlinear thermal expansion is calculated using the balanced heat flux products (so no expansion due to net warming, see section 3.6) and estimated to be 1 - 6 mm year$^{-1}$. Hence, both the magnitude of this effect as well as the difference between

the products, is of the same order as observed GMSL rise. The differences are related to different emphasize between products of where heat leaves or enters the ocean, stating the importance of carefully constructing heat flux products. These results are comparable to, albeit a bit smaller, that found by Griffies and Greatbatch (2012) (their Fig. 7). Note that the impact from both the nonlinear haline contraction and mass fluxes are about an order of magnitude smaller than that from nonlinear thermal expansion (Table 1). In section 4.6 the balance between densification upon mixing and nonlinear thermal and haline expansion

is examined.

### 4.4   GMSL rise due to different SWR penetration parameterizations

The total incoming SWR at the surface is vertically redistributed according to some vertical structure function $F(z)$ (section 4.4). This means that a part of the heat reaching the surface, is actually accumulating and transforming water below the surface. Most often it will be cooler there with a smaller thermal expansion coefficient $\alpha$, and thus a net smaller volume

increase compared to the hypothetical situation in which all SWR is absorbed at the surface. The results in Table 1 used the vertical distribution function of Morel and Antoine (1994) (Eq. 18). When the other two specified functions (section 4.4) are applied to the balanced products for CORE, the observed change in GMSL level due to the total heat flux is 4.7, 5.8, and 6.7 mm year$^{-1}$ for $F_{PS77}$, $F_{MA94}$ (as above) and $F_{SF}$, respectively. Taking the balanced products for OA, this gives -0.05, 1.0 and 2.0 mm year$^{-1}$ for $F_{PS77}$, $F_{MA94}$ (as above) and $F_{SF}$, respectively. As $F_{PS77}$ allows for the deepest penetration, this will

lead to the least increase, followed by $F_{MA94}$ and $F_{SF}$. For the unbalanced products the difference are similar (not shown). In conclusion, the choice of SWR depth penetration parameterization an change GMSL rise estimates with a magnitude of order 1.0 mm year$^{-1}$.

### 4.5   GMSL rise due to geothermal heating

Geothermal heat injection from the sea floor leads to sea level expansions and predominantly occurs along ocean ridges, with

additional hotspots in the Caribbean Sea and the waters around South East Asia (Fig. 5h). The great ocean basins have values

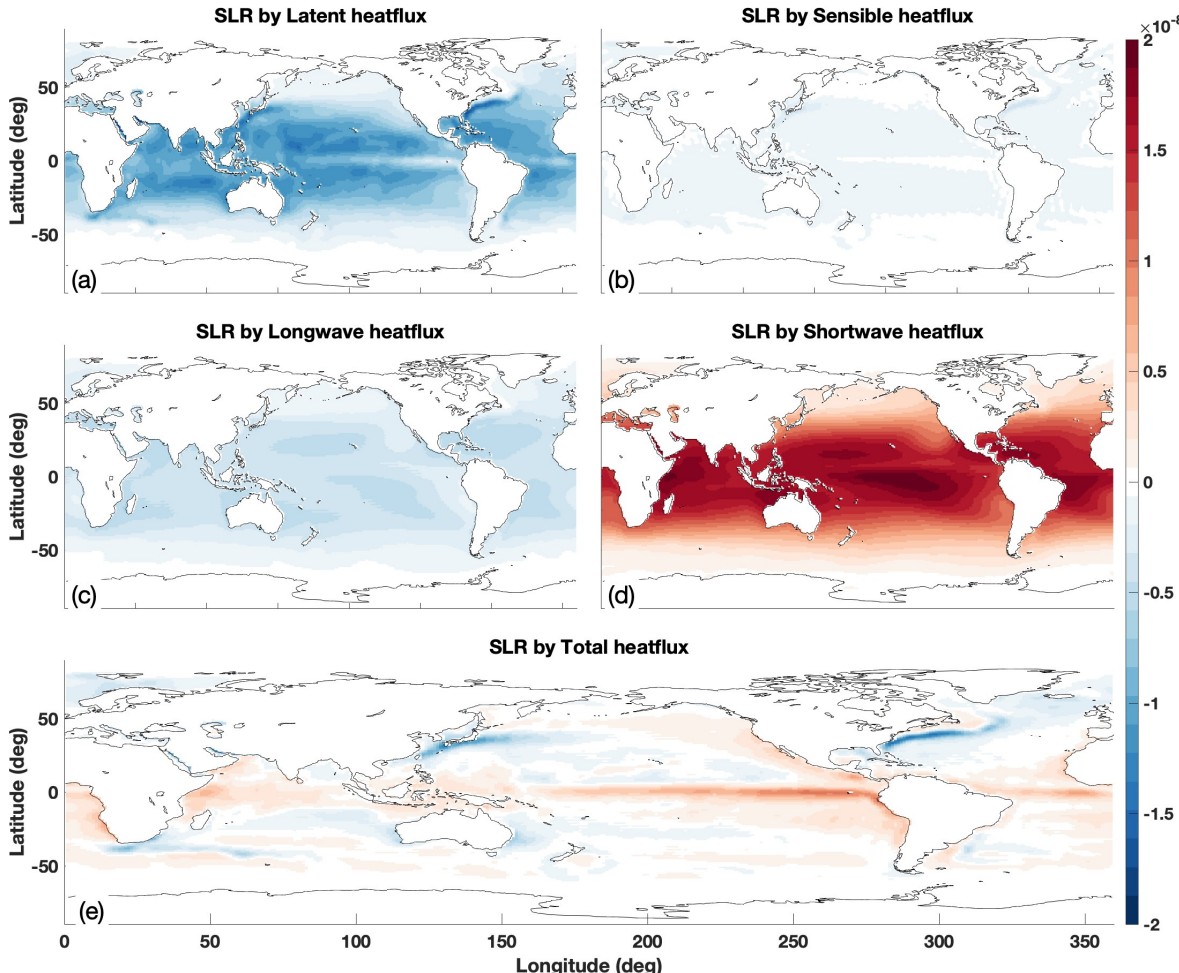

**Figure 3.** The spatial impact of heat fluxes on sea level rise (m s$^{-1}$) due to latent heat flux (a), sensible heat flux (b), longwave heat flux (c), shortwave heat flux (d) and their sum (e). Derived and discussed in sections 2.2, 3.5, 4.3, Eq. 5, and Table 1. For penetration of SWR the Morel and Antoine (1994) parameterization is used (Eq. 18).



of an order of magnitude lower, but cover large surface areas. The total impact of geothermal heating on GMSL is 0.08 mm year$^{-1}$. This is relatively small compared to other processes (Table 5) and is exactly similar to that calculated in a numerical model (Griffies and Greatbatch, 2012) (their Table 1).

### 4.6  GMSL rise due to mixing

Here the impact on GMSL due to mixing processes are quantified (Section 2.3, Eq. 6). The impact on GMSL rise by both constant diffusivities as well as the more realistic spatially varying diffusivities (section 3) are compared and contrasted. The results for a constant diffusivity are simply proportional to the diffusivity, and can therefore easily be rescaled with a different constant diffusivity in mind.

For variable diffusivities, the largest impact on GMSL is by vertical cabbeling (-2 mm year$^{-1}$), and to a lesser extend by
horizontal (-0.4 mm year$^{-1}$) and neutral cabbeling (-0.5 mm year$^{-1}$). Together decreasing GMSL with a rate of about -3 mm year$^{-1}$ (Table 2). Somewhat important are vertical thermobaricity (0.2 mm year$^{-1}$) and the bottom boundary condition term (-0.1 mm year$^{-1}$). All other mixing-related terms have an almost negligible impact on GMSL rise (Table 2).

Griffies and Greatbatch (2012) found that the KPP-mixing scheme for the surface boundary layer (Large et al., 1994; Van Roekel et al., 2018) has a comparable impact on GMSL rise as vertical diffusion over the entire ocean. The small-scale
diffusivities used in this study (de Lavergne et al., 2019), don't include a surface boundary layer parameterizations and are known to underestimated by the diffusivities in that region. The impact of vertical mixing could therefore be underestimated by about a factor 2. The use of a constant diffusivity is an overly simplistic alternative that instead might overestimate the impact of vertical mixing and results in $D_{\text{constant}} > D(x,y,z)$ in the upper 2500 meter of the ocean. This explains the difference of a factor 5 between in GMSL rise due to vertical cabbeling when using a variable or constant vertical diffusivity $D$. The
best way to narrow down this estimate is to first include an observational-based estimate of a surface layer boundary mixing scheme, which is beyond the scope of this study.

Taken together, the total impact of all mixing on GMSL rise is between -3 and -11 mm year. This indicates that both the impact of mixing itself, as well as the uncertainty between mixing parameterisations, are of same order as the observed GMSL rise. In addition, the impact of mixing is comparable in magnitude and of opposite sign to that by nonlinear thermal expansion
(Table 1, section 4.3), suggesting that densification upon mixing will keep the ocean from ever expanding.

Vertical cabbeling takes place where there is significant vertical mixing by internal waves (de Lavergne et al., 2020) in combination with vertical gradients of temperature and salinity (Eq. C22), which is mostly around topography (Fig. 4a). Although the overall impact of vertical diffusion on GMSL rise is comparable to that found in Griffies and Greatbatch (2012), their spatial structure of this effect is very different (their figure 10a). The differences between these figures is mostly explained
by means of the different diffusivity parameterisations used. Vertical thermobaricity is of opposite sign to cabbeling and smaller, increasing GMSL at locations where vertical cabbeling is also strong.

Neutral temperature and salinity gradients are particularly strong in the Southern Ocean at mid-depths, near western boundary currents and to a lesser extend in the greater ocean basins (Groeskamp et al., 2019a) The mesoscale diffusivity used, are particularly strong near western boundary currents and in some of the subtropical regions (Groeskamp et al., 2020). Together





| Process: | Variable Diffusion | Constant Diffusion |
|---|---|---|
| Total (unit of mm year$^{-1}$) | **-2.8** | **-11** |
| Cabbeling Vertical $P_{C_b}^{verr}$ | -2.0 | -10 |
| Cabbeling Horizontal $P_{C_b}^{hor}$ | -0.48 | -0.75 |
| Cabbeling Neutral $P_{C_b}^{ntr}$ | -0.44 | -0.41 |
| Thermobaricity Vertical $P_{T_b}^{ver}$ | 0.16 | 0.49 |
| Bottom Bdy-term $\mathbf{R}^{hor} \cdot \nabla_H(-H)$ | -0.12 | -9.0 x $10^{-2}$ |
| Thermobaricity Neutral $P_{T_b}^{ntr}$ | -3.7 x $10^{-2}$ | -2.7 x $10^{-2}$ |
| $D^{-1} \|\mathbf{R}^{ver}\|^2$-term | -1.4 x $10^{-2}$ | -9.5 x $10^{-2}$ |
| $K_H^{-1} \|\mathbf{R}^{hor}\|^2$-term | -3.3 x $10^{-3}$ | -2.9 x $10^{-3}$ |
| $\rho g \kappa \mathbf{k} \cdot \mathbf{R}^{ver}$-term | -6.0 x $10^{-3}$ | -2.3 x $10^{-2}$ |
| Surface Bdy-term $\mathbf{R}^{hor} \cdot \nabla_H(\eta)$ | -2.5 x $10^{-5}$ | -3.7 x $10^{-5}$ |
| Thermobaricity Horizontal $P_{T_b}^{hor}$ | 1.5 x $10^{-5}$ | 2.2 x $10^{-5}$ |

**Table 2.** Area weighted GMSL rise in mm year$^{-1}$, calculated using Eq. 2 for different mixing terms as described in section 2.3 and Eq. 6 and shown in Figs. 4 and 5.

this means the impact of cabbeling on GMSL rise is centered around western boundary currents and to some extend in the Southern Ocean (Fig. 4c). Location comparable to other studies considering cabbeling in the ocean (Groeskamp et al., 2016; Klocker and McDougall, 2010; Griffies and Greatbatch, 2012; Stewart et al., 2017; Nycander et al., 2015). Horizontal cabbeling is defined using the same diffusivity as neutral mixing, but only in the mixed layer. This means it is more pronounced where mixed layers are also deeper (Fig. 4b), such as in the Southern Ocean and the North Atlantic (de Boyer Montégut et al., 2004)

The total spatial impact of mixing on GMSL rise shows global decreasing sea level (Fig. 4e), with the highest values where the different cabbeling processes are significant. The spatial distribution of the small (negligible) terms are shown for completeness 5), but not further discussed.

### 4.7   GMSL rise due to stirring

The range of the impact of stirring on sea level (section 2.4, Eq. 11) is investigated by comparing results using a constant

mesoscale diffusivity to that using a spatially varying mesoscale diffusivity (section 3). There are only three terms contributing to stirring, of which the first term is zero after global integration (Eq. 11). Of the remaining two terms, it is only the "production term" $P_{stir}$ that has a significant impact on GMSL rise of about -0.4 to -0.6 mm year$^{-1}$ (Table 3). This is non negligible, but still about 8 times smaller than observed GMSL rise.

    The stirring parameterization includes a factor $K \mathbf{S}$ (Eq. D5), causing its main effect to be where the combination of both

neutral slopes and mesoscale diffusivity are large. This is between 40°S an 50°S and in western boundary currents (Fig. 6). Although its global mean impact is moderate, it could locally be important.





**Figure 4.** The spatial impact on sea level rise (m s$^{-1}$) by the most important mixing terms, due to vertical cabbeling (a), horizontal cabbeling (b), neutral cabbeling (c), vertical thermobaricity (d) and the sum of all mixing terms (e), including the ones shown in Fig. 5. The mixing terms are derived and discussed in sections 2.3, 3.2, 4.6, Eq. 6, and Table 2.

| Process | Variable Diffusion mm year$^{-1}$ | Constant Diffusion mm year$^{-1}$ |
|---|---|---|
| Total | **-0.55** | **-0.36** |
| Production by Stirring $P_{stir}$ | -0.53 | -0.35 |
| $\mathbf{R}^{stir} \cdot \nabla \ln \rho$-term | -1.4 x 10$^{-2}$ | -9.1 x 10$^{-3}$ |

**Table 3.** Area weighted GMSL rise in mm year$^{-1}$, calculated using Eq. 2 for stirring. See also sections 2.4, 4.7, Eq. 11 and Fig. 6.



**Figure 5.** The spatial impact on sea level rise (m s$^{-1}$) by geothermal heating and the mixing terms with a small impact. Shown are the bottom boundary term (a), neutral thermobaricity (b), the diffusion-density interaction with vertical mixing $D^{-1}\left|\mathbf{R}^{\mathrm{ver}}\right|^2$-term (c), the diffusion-density interaction with horizontal mixing $K_{\mathrm{H}}^{-1}\left|\mathbf{R}^{\mathrm{hor}}\right|^2$-term (d), the horizontal Thermobaricity (e), the the diffusion-density interaction with horizontal mixing $\rho g \kappa \mathbf{k} \cdot \mathbf{R}^{\mathrm{ver}}$-term (f), the surface Boundary term $\mathbf{R}^{\mathrm{hor}} \cdot \nabla_{\mathrm{H}}(\eta)$ (g), and the geothermal heating (h). Note the different the color bars. The mixing terms (a-g) are derived and discussed in sections 2.3, 3.2, 4.6, Eq. 6, and Table 2. The geothermal heat (h) is derived and discussed in sections 2.2, 3.5, 4.5, Eq. 5, and Table 5.



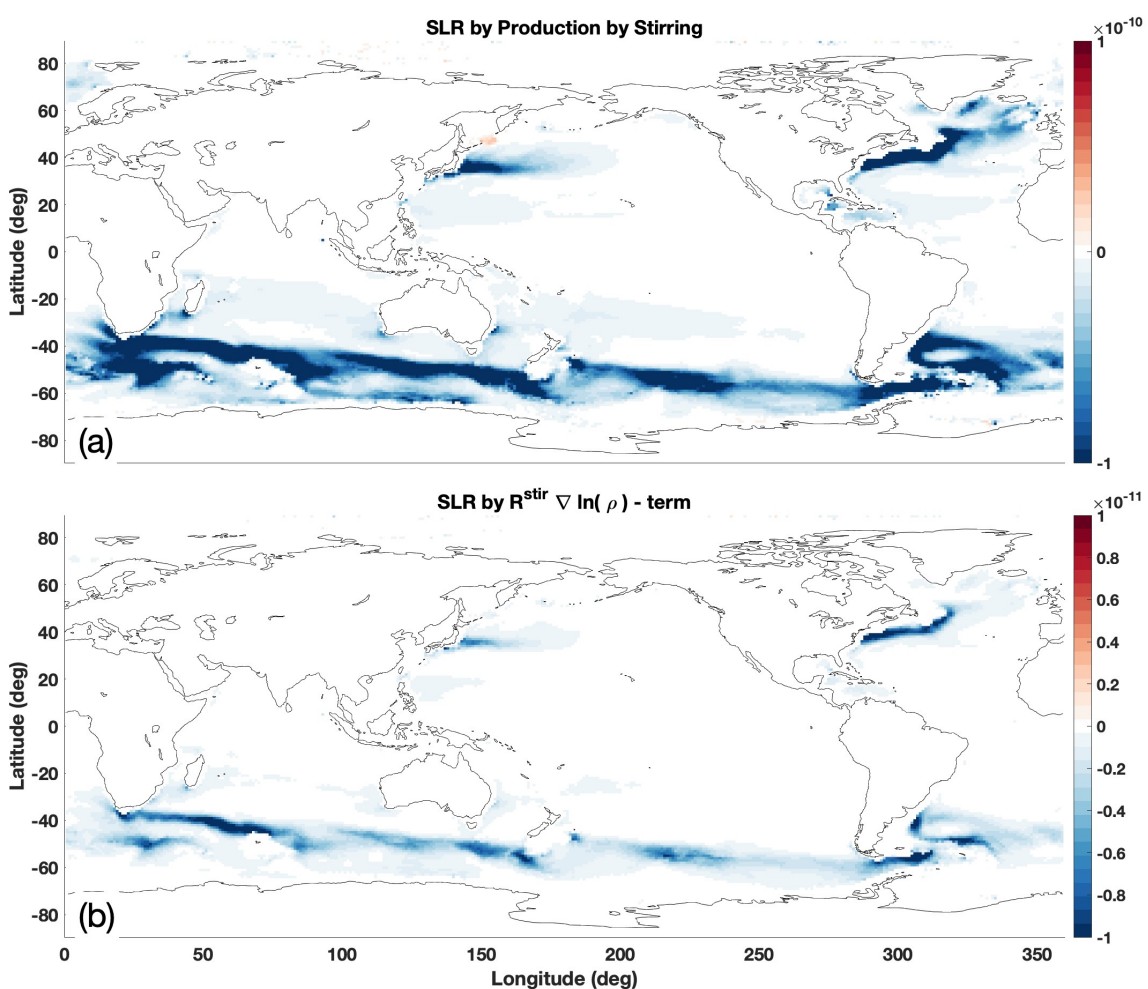

**Figure 6.** The spatial impact of the stirring terms on sea level rise (m s$^{-1}$), due to stirring production term (a) and the $\mathbf{R}^{\mathrm{stir}} \cdot \nabla \ln \rho$-term (b). Note the different scales of the colorbars. The stirring terms are derived and discussed in sections 2.4, 3.2, 4.7, Eq. 11, and Table 3.





| Process | Variable Diffusion mm year$^{-1}$ | Constant Diffusion mm year$^{-1}$ |
|---|---|---|
| Total | **-2.7** | **-2.4** |
| $\mathrm{P}^{\mathrm{ntr}} - \mathrm{P}^{\mathrm{(perfectly\ neutral)}}$ | -2.5 | -2.2 |
| $\frac{\partial \eta}{\partial t}\big\|_{\mathrm{stirring}}^{\mathrm{non-neutral}}$-term | -0.16 | -0.13 |
| Bottom Bdy-term $\mathbf{R}^{\mathrm{ntr}} \cdot \nabla_{\mathrm{H}}(-H)$ | -7.4 x 10$^{-4}$ | -4.4 x 10$^{-2}$ |
| $\kappa \nabla_{\mathrm{N}} P \cdot \mathbf{R}^{\mathrm{ntr}}$-term | 1.8 x 10$^{-5}$ | 6.4 x 10$^{-5}$ |
| $K_{\mathrm{N}}^{-1} \left| \mathbf{R}^{\mathrm{ntr}} \right|^2$-term | -1.1 x 10$^{-6}$ | -5.5 x 10$^{-6}$ |

**Table 4.** Area weighted GMSL rise in mm year$^{-1}$, calculated using Eq. 2 for non-neutral terms. See also sections 2.5, 4.8, Eq. 14 and Fig. 7.

## 4.8 GMSL rise due to Non-Neutrality

Here the impact of non-neutrality on GMSL is quantified (Section 2.5, Eq. 14). As explained in section 3, two methods are used for calculating neutral slopes and gradients. The results from the VENM algorithm are used as "perfectly neutral" in Eq.

14, while using the results from the "local" method as "ntr". As VENM is not perfectly neutral, the results should be interpret as the order of magnitude of the improvement that can be made when accurate neutral physics is implemented.

The first (non-divergent) term in Eq. 14 has no net contribution to GMSL rise by construction and is discussed in section 4.9. The three non-neutral terms that are related to $\mathbf{R}^{\mathrm{ntr}}$ (Eq. 14), all have very small contribution to GMSL (Table 4, Fig. 7b,c,d). Note these can be directly calculated using the gradients obtained from the VENM method, and these terms would be larger

when using the "local" method as that is much less neutral and more irregular (Groeskamp et al., 2019a). However, these terms are small and don't make a significant contribution, even with the "local" method, and are not further discus.

The main impact of non-neutrality to GMSL rise comes from the neutral cabbeling and thermobaricity terms (-2.5 mm year$^{-1}$), and to a lesser extend from eddy stirring (-0.2 mm year$^{-1}$). The impact of calculating these terms using different method is in total about -3 mm year$^{-1}$, with a small impact from difference in diffusivity. This means that the use of the "local"

method, makes an error of at least the same order of magnitude as observed GMSL rise rates themselves.

## 4.9 GMSL redistribution terms

The four redistribution terms that have local impact on sea level, but no net impact on GMSL, are here discussed. These four terms are due to ocean currents (Eq. 10, section 2.4), from horizontal diffusion (Eq. 6, section 2.3), from neutral diffusion (a non-neutrality term, Eq. 14, section 2.5) and from stirring (Eq. 11, section 2.4). In general the redistribution terms are large

(except the neutral term) and thus have significant local impact on sea level. By far the largest impact is due to dynamical sea level changes, which is comparable to vertically integrated vertical velocity of Fig. 4 of Griffies and Greatbatch (2012) (except for a factor $\rho$). In all cases, clear patterns of positive and negative due to the divergence operator exist (Fig. 8). The shape and location of these patterns are related to specific processes. For example; the horizontal diffusion redistributes volume from





**Figure 7.** The spatial impact of the non-neutral terms on sea level rise (m s$^{-1}$), due to the production terms (a), the stirring production term (b), the $K_N^{-1} \left| \mathbf{R}^{\text{ntr}} \right|^2$-term (c), the bottom boundary term (d), the $\kappa \nabla_N P \cdot \mathbf{R}^{\text{ntr}}$-term (e), and the sum of all these terms (f). Note the different scales of the color bars. The non-neutral terms are derived and discussed in sections 2.5, 3.3, 4.8, Eq. 14, and Table 4.

**Figure 8.** The spatial impact of the redistribution terms on sea level rise (m s$^{-1}$), due to horizontal divergence (a), non-neutral divergence (b), stirring divergence (c) and ocean dynamics (d). Note the different scales of the color bars. The divergence terms are discussed in section 4.9.

the subtropics to the equator, while stirring does this within western boundary currents. Overall these results are not further
examined, but the figures are provided for completeness.

## 5 Summary and Discussion

Observational based mass and heat fluxes are by far the largest contribution to GMSL changes, but are notoriously hard to balance due to a lack of observational constraints. For example, the GMSL rise estimates based on the mass flux (-35 to



| Process | mm year$^{-1}$ | |
|---|---|---|
| Mass [balanced product] | -35 to 3 | [0.05 to 0.1] |
| Heat [balanced product] | 13 to 57 | [1 to 6] |
| Fresh water [balanced product] | -0.2 to -1.2 | [-0.3] |
| Diffusion | -11 to -3 | |
| Stirring | -0.6 to -0.4 | |
| Non-neutrality | -3 | |
| Shortwave radiation parameterisation | $\pm$ 1 | |
| Geothermal Heating | 0.08 | |

**Table 5.** Summary of the area weighted GMSL rise in mm year$^{-1}$ for the different processes discussed in this paper. The results from the balances mass and heat flux product are given in brackets.

3 mm year$^{-1}$, Table 5) or the heat flux (13 to 57 mm year$^{-1}$, Table 5) are both about 10 times larger than the observed

GMSL rise. For comparison, Griffies and Greatbatch (2012) found the net impact of the ocean mass flux on sea level to be 0.8 mm year$^{-1}$ in a numerical model environment (their Fig 2). In addition, it shows that the differences between the GMSL rise estimates calculated using different heat and mass fluxes, are also 10 times larger than the observed GMSL rise. With such large inaccuracies in estimating GMSL rise, it is not yet possible to close the GMSL budget using the sum of the parts (bottom-up approach).

Therefore this study also uses heat and mass flux products that are artificially made to have a global integrated net-zero mass and heat flux (Appendix E). This approach is somewhat comparable to numerical ocean models that remove global mass imbalances at each time step in order to remove long term drift (Griffies, 2012). For the balanced mass fluxes, the impact on GMSL is about 0.1 mm year$^{-1}$. The residual is a consequence of nonlinear weighting of the mass fluxes by the ocean surface density (Eq. 4). The mass flux can also be recalculated into an equivalent salt flux (using a factor $\beta S_A$, Eq. B5) that affect

density and changes GMSL with about -0.3 mm year$^{-1}$.

   The impact of balanced heat flux products on GMSL is reduced by an order of magnitude compared to the unbalanced products to 1-6 mm year$^{-1}$. This is not due to "regular" thermal expansion from climate change warming, as there is net-zero heat going into the ocean. Instead this is due to nonlinear thermal expansion as a consequence of the heat fluxes being weighted by the thermal expansion coefficient that varies strongly with temperature, before globally integrated (Eq. B5). Note

that (slight) differences in spatial variation between the two heat flux products, lead to a significant difference in GMSL rise estimate of 5 mm year$^{-1}$. Hence, both the value and the uncertainty in calculating the nonlinear thermal expansion have a first order impact on GMSL budget.

   The combination of horizontal, neutral and vertical mixing leads to densification upon mixing and subsequent reduction in GMSL of -3 to -11 mm year$^{-1}$. This is mostly due to vertical mixing, and to a lesser extend due to horizontal and neutral

mixing. Note that thermobaricity, a nonlinear effect related to pressure and temperature changes, can lead to both a sea level rise and fall. However, this term is in general an order of magnitude smaller than the impact of densification upon mixing.





The range of the impact of mixing on GMSL rise depends mostly on the parameterizations used for mixing diffusivity (mixing strength). I conclude that 1) mixing itself has a first order effect on GMSL rise, and 2) that the difference between mixing strength parameterization have an impact on GMSL that is of the same order as the observed GMSL rise. In short, mixing
matters for GMSL, which is of interest for numerical modeling purposes because mixing parameterizations can vary strongly between models (Pradal and Gnanadesikan, 2014), therewith differently impacting GMSL rise budgets. Stirring of heat and salt by mesoscale eddies decrease GMSL rise with a rate of -0.4 to -0.6 mm year$^{-1}$, which is of importance, but smaller than the impact of diffusion.

In this study both nonlinear thermal expansion and densification upon mixing are separately quantified and are, within the
range of the estimates, indeed of the same order of magnitude and of opposite sign. Note that, if the ocean would be of uniform temperature and salinity, both effects would not exist. Hence these two processes will always go hand in hand, because as heat fluxes create extremes and therewith nonlinear thermal expansion, mixing will work to make the ocean uniform and therewith cause densification upon mixing. As models include choices about mixing parameterizations and boundary flux products, this leads to some balance between nonlinear thermal expansion and densification upon mixing. It is complicated to understand
what impact these choices have on the GMSL budgets and related predictions of future sea level rise. In addition, the time scales related to the impact of densification upon mixing or nonlinear thermal expansion are likely very different, meaning that there is also some time lag between the impact of these two processes. Understanding this exact interplay is of interest for future work, but beyond the scope of this study. This study thus emphasizes the importance of ocean mixing, not only for circulation (de Lavergne et al., 2022), climate (Melet et al., 2022) and biogechemical processes (Lévy et al., 2022; Spingys
et al., 2021), but also for sea level rise (Gille, 2004; Jayne et al., 2004).

This study shows that the way neutral physics is implemented, has a first order effect on GMSL rise. Incorrect implementation of neutral physics leads to additional mixing and related densification upon mixing. Differences in neutral physics methods to calculate neutral slopes and gradients lead to a GMSL difference of about -3 mm year$^{-1}$, which is a leading order term. This leads to a low bias, something also hinted at by Gille (2004) in a numerical model. This emphasizes the need to integrate the
most advanced and accurate methods for neutral physics calculation in numerical models for predicting future sea level rise (Groeskamp et al., 2019a; Shao et al., 2020).

Different ways to parameterize the vertical distribution of shortwave radiation, causes differences in which water is heated. Parameterization that allow for deeper penetration, heat colder waters and thus have reduced impact on sea level rise, compared to parameterizations that allow for less deep penetration. Differences in sea level rise between such parameterizations are about
$\pm 1$ mm year$^{-1}$. Not only is this a leading order term, the effect is consistent over time and has the potential to accumulate to 10 cm per difference per century differences, between climate models.

The impact of pressure variations on density ($\kappa \frac{DP}{Dt}$, last term, Eq. A1) are not accounted for. This is rationalized because 1) the impact is measurable but small (Dewar et al., 1998) , and 2) Griffies and Greatbatch (2012) showed the impact on GMSL rise of this term is about 1000-10.000 times smaller than recent sea level rise estimates. As this term is difficult to calculate
from observational based products and almost negligible, it is not further investigated in this study.





## 6 Conclusions

By integrating over the ocean hydrography, using satellite altimetry and ice sheet modeling, the GMSL budget can be accurately closed (Ludwigsen et al., 2024). However, this top-down approach glosses over the impact of individual (ocean) physical processes on the GMSL budget. This study therefore applies a bottom-up approach in which the contribution of individual

physical processes contributing to GMSL rise are estimated from observational based products. The focus is particularly on the impact of diffusion, stirring, neutral physics, shortwave radiation and boundary fluxes, that all change oceanic density and therewith GMSL. For completeness, but not as a key focus, direct mass fluxes and the impact of ocean currents are also examined. It is valuable to be able to close the the GMSL rise budget as estimated from observations by summing up the contribution from the physical processes underlying changes in the GMSL rise budget. This provides understanding in the

fundamental processes behind the observed global sea level rise and how these processes may change and in a transient ocean and climate.

This study provides a comparison of both the magnitude and uncertainty of the impact on GMSL by single processes or parameterizations. With the observed GMSL rise currently being about 4 mm year$^{-1}$, this indicates that processes causing changes of the order of 1 mm year$^{-1}$ can be considered leading order terms in calculating GMSL rise. For accurately closing

the GMSL rise budget, one should arguably have an accuracy of about 0.1 mm year$^{-1}$. This immediately clarifies that it is not yet possible to close the GMSL budget by means of the bottom-up approach applied in this study. For example, there are large difference between mass and heat flux products, partly due to a lack of observational based constraints. This leads to differences in GMSL rise estimates that are about 40 mm year$^{-1}$ between boundary heat or mass flux products (sections 4.1 , 4.3 and Table 5). This is about two orders of magnitude larger than the accuracy required to close the budget. Therefore this

study also used artificially balanced (i.e., globally net-zero) boundary mass and heat flux products for calculating the impacts on GMSL rise. Taken together, this study then finds that:

 – One can't close the GMSL rise budget using unbalanced heat or mass flux products (sections 4.1, 4.3)

 – Even between different balanced heat flux products, the resulting GMSL rise estimates difference are of the same order as observed GMSL rise itself. This indicates that the spatial distributed of a heat flux product plays an important role for

the GMSL rise budget (sections 4.1, 4.3).

 – Mixing strength parameterizations have a leading order impact on GMSL rise estimates (section 4.6).

 – Implementation of neutral physics has a leading order impact on GMSL rise estimates (section 4.8).

 – The choice for shortwave radiation parameterisations has has a leading order impact on GMSL rise budgets, and its impact accumulates over time (section 4.4).

– Parameterized eddy advection and freshwater fluxes have a second order impact on GMSL (sections 4.7 and 4.2).




- Nonlinear thermal expansion and densification upon mixing go hand in hand, are of the same order of magnitude and of opposite sign, therewith compensating each other. Albeit over different time scales, due to different physical processes and at different locations (section 5).

The accuracy of the estimates are limited by both a lack of knowledge and observational based constraints for several of the physical processes involved (e.g. boundary heat and mass fluxes, mixing, Shortwave radiation), as well as due to the complexity to numerically implementing neutral physics. The above points should also be of interest to ocean modelers, as they have specific choices to make about which method they choose to represent heat fluxes, mixing diffusivities, shortwave penetration, eddy stirring and neutral physics. All these factors significantly impact on GMSL rise calculations. It remains unclear how the combination of these choices would impact GMSL rise prediction in for example IPCC-class models, as they

require significant spin up and equilibrium time in which some of these errors might balance out, or may lead to the right estimate for the wrong reason. Therefore, these results advocate for a thorough analyses of these processes in both models and observations, to increase understanding of such choices on GMSL rise predictions and increase the our accuracy of predicted future sea level rise upon which policy will be based.

## Appendix A: The material derivative of density

In Eq. 1, the material derivative of density is given by:

$$\frac{\mathrm{d}\rho}{\mathrm{d}t} = -\alpha\rho\frac{\mathrm{d}\Theta}{\mathrm{d}t} + \beta\rho\frac{\mathrm{d}S_\mathrm{A}}{\mathrm{d}t} + \kappa\rho\frac{\mathrm{d}P}{\mathrm{d}t}. \tag{A1}$$

Here $\frac{\mathrm{d}\Theta}{\mathrm{d}t}$, $\frac{\mathrm{d}S_\mathrm{A}}{\mathrm{d}t}$ and $\frac{\mathrm{d}P}{\mathrm{d}t}$ are the material derivatives of Conservative Temperature $\Theta$ (McDougall, 2003; Graham and McDougall, 2013), Absolute Salinity $S_\mathrm{A}$ (McDougall et al., 2012; IOC et al., 2010) and pressure $P$, respectively. Here $\alpha$ is the thermal expansion coefficient, $\beta$ is the saline contraction coefficient and $\kappa$ is the isentropic compressible respectively, given by:

$$\alpha = -\frac{1}{\rho}\frac{\partial\rho}{\partial\Theta}|_{S_\mathrm{A},P}, \quad \beta = \frac{1}{\rho}\frac{\partial\rho}{\partial S_\mathrm{A}}|_{\Theta,P}, \quad \kappa = \frac{1}{\rho}\frac{\partial\rho}{\partial P}|_{S_\mathrm{A},\Theta}. \tag{A2}$$

Here the $|_{S_\mathrm{A},P}$ indicates that the derivative is obtained at constant $S_\mathrm{A}$ and $P$, etc. Changes in density are related to changes in $S_\mathrm{A}$, $\Theta$ and $P$ through $\alpha$, $\beta$ and $\kappa$. Because $\alpha$, $\beta$ and $\kappa$ also depend on $S_\mathrm{A}$, $\Theta$ and $P$, the equation of state is nonlinear. The convergence of heat and salt that can be written as:

$$\rho\frac{\mathrm{d}\Theta}{\mathrm{d}t} = -\nabla\cdot\left(\mathbf{J}^\Theta_\mathrm{diff} + \mathbf{J}^\Theta_\mathrm{stir}\right) + F^\Theta_\mathrm{mass} + \rho F^\Theta_\mathrm{source}, \tag{A3}$$

$$\rho\frac{\mathrm{d}S_\mathrm{A}}{\mathrm{d}t} = -\nabla\cdot\left(\mathbf{J}^{S_\mathrm{A}}_\mathrm{diff} + \mathbf{J}^{S_\mathrm{A}}_\mathrm{stir}\right) + F^{S_\mathrm{A}}_\mathrm{mass} + \rho F^{S_\mathrm{A}}_\mathrm{source}. \tag{A4}$$

Here salt and heat changes are due to convergence of diffusive fluxes are given by $\mathbf{J}^\Theta_\mathrm{diff}$ and $\mathbf{J}^{S_\mathrm{A}}_\mathrm{diff}$ (in tracer kg m$^{-2}$ s$^{-1}$), and include a diversity of mixing processes that are detailed in section 2.3. The minus sign assures positive numbers when heat or salt accumulate (converge). Salt and heat convergence due to advective subgridscale processes or 'skew fluxes' are given by $\mathbf{J}^\Theta_\mathrm{stir}$ and $\mathbf{J}^{S_\mathrm{A}}_\mathrm{stir}$ (in tracer kg m$^{-2}$ s$^{-1}$) (Gent and McWilliams, 1990; Gent et al., 1995; Griffies, 1998; McDougall and McIntosh,

2001). The changes of heat and salt due to boundary mass fluxes are given by $F^\Theta_\mathrm{mass}$ and $F^{S_\mathrm{A}}_\mathrm{mass}$. Meanwhile the source terms



$F^{S_A}_{\text{source}}$ and $F^{\Theta}_{\text{source}}$ (tracer kg m$^{-3}$ s$^{-1}$) contain all other possible direct sources and sinks of salt and heat. Calculations of density changes due to pressure (last term, Eq. A1) is not further investigated in this study (see section 5)

**Appendix B: Specifying the impact of boundary fluxes of salinity and temperature on density and sea level.**

In this section the impact of mass and source fluxes of salinity and temperature are combined, to express their impact on density

and sea level. Surface mass fluxes $Q_{\text{mass}}$ change salinity and temperature through $F^{S_A}_{\text{mass}}$ and $F^{\Theta}_{\text{mass}}$ in Eq. A4. Combined, they alter density as follows:

$$F^{\rho}_{\text{mass}} \quad = \quad \underbrace{-\alpha Q_{\text{mass}}\left(\Theta_{\text{m}} - \Theta\right)\delta\left(z - \eta\right)}_{F^{\Theta}_{\text{mass}}} + \underbrace{\beta\, Q_{\text{mass}}\,\left(S_{\text{m}} - S_{\text{A}}\right)\delta\left(z - \eta\right)}_{F^{S_A}_{\text{mass}}}. \tag{B1}$$

Here $F^{\rho}_{\text{mass}}$ (kg m$^{-3}$ s$^{-1}$) contains $\Theta_{\text{m}}$ and $S_{\text{m}}$ that are the mass-flux-weighted average of the salinity and temperature of the various components of the mass flux that are entering the ocean, while $\Theta$ and $S_{\text{A}}$ are the oceanic values at the point of entry. The Dirac delta function $\delta\left(z - \eta\right)$ has units of inverse length (m$^{-1}$). Note that it is often assumed that the temperature of the

mass flux equals the ocean such that $(\Theta_{\text{m}} - \Theta) = 0$, while the air–sea mass flux generally has a vanishing salinity ($S_{\text{m}} = 0$), making the salinity term an important term in the sea level budget (Nurser and Griffies, 2019).

Direct sources of salinity and heat at the surface of the ocean also impact the density budget. Surface heat fluxes ($F^{\Theta}_{\text{surface}}$, W m$^{-2}$) are given by longwave radiation as well as turbulent fluxes associated with latent and sensible heat. Surface salt fluxes ($F^{S}_{\text{surface}}$, g m$^{-2}$ s$^{-1}$) are associated with for example sea ice or spray. This term can often be ignored, as is done in this study.

The impact of these fluxes gives a change in density according to:

$$F^{\rho}_{\text{surface}} \quad = \quad \left(-\frac{\alpha}{C_p}F^{\Theta}_{\text{surface}} + \beta F^{S}_{\text{surface}}\right)\delta\left(z - \eta\right). \tag{B2}$$

Here $F^{\rho}_{\text{surface}}$ is given in kg m$^{-3}$ s$^{-1}$ and $C_p$ is the seawater heat capacity (J kg$^{-1}$ K$^{-1}$). At the ocean bottom, geothermal heating ($F^{\Theta}_{\text{geo}}$, W m$^{-2}$) is a direct source of heat that alters the density budget as:

$$F^{\rho}_{\text{geo}} \quad = \quad -\frac{\alpha}{C_p}F^{\Theta}_{\text{geo}}\delta\left(z + H\right). \tag{B3}$$

Here $F^{\rho}_{\text{geo}}$ is given in kg m$^{-3}$ s$^{-1}$. Shortwave radiation (SWR) is a direct source of heat that enters the ocean at the surface and penetrates to deeper layers depending on the clarity of the water (Paulson and Simpson, 1977). The impact on density by

convergence of SWR is given by:

$$F^{\rho}_{\text{swr}} \quad = \quad \left(\frac{\alpha}{C_p}\nabla\cdot\mathbf{J}^{\text{swr}}_Q\right) = \frac{\alpha}{C_p}Q_{\text{swr}}\frac{\partial F(z)}{\partial z}. \tag{B4}$$

Here $F^{\rho}_{\text{swr}}$ is given in kg m$^{-3}$ s$^{-1}$ and $\mathbf{J}^{\text{swr}}_Q = Q_{\text{swr}}F(z)\mathbf{k}$ is the amount of SWR at the surface ($Q_{\text{swr}}$, W m$^{-2}$) spread over depth according to the function $F(z)$. The convergence of this depth-depending influx leads to a net heating (hence the extra minus sign to assure positive convergence). Note that $F(z)$ can depend also on factors such as water clarity and chlorophyll concentrations (Lewis et al., 1990; Morel and Antoine, 1994; Ohlmann, 2003). Accounting for all the above and inserting that



into Eq. 1, and using that $(\Theta_{\mathrm{m}} - \Theta) = 0$, $S_{\mathrm{m}} = 0$, $F_{\mathrm{source}}^{S_{\mathrm{A}}} = 0$ to find:

$$
\begin{aligned}
\left. \frac{\partial \eta}{\partial t} \right|_{\mathrm{boundary}} &= -\int_{H}^{\eta} \frac{1}{\rho} \left( F_{\mathrm{mass}}^{\rho} + F_{\mathrm{surface}}^{\rho} + F_{\mathrm{geo}}^{\rho} + F_{\mathrm{swr}}^{\rho} \right) dz \\
&= \left[ \frac{\beta S_{\mathrm{A}}}{\rho} Q_{\mathrm{mass}} + \frac{\alpha}{\rho C_p} F_{\mathrm{surface}}^{\Theta} \right]_{\eta} + \left[ \frac{\alpha}{\rho C_p} F_{\mathrm{geo}}^{\Theta} \right]_{-H} \\
&\quad - \int_{H}^{\eta} \frac{\alpha}{\rho C_p} Q_{\mathrm{swr}} \frac{\partial f(z)}{\partial z} dz.
\end{aligned}
\tag{B5}
$$

Note that in this integral, even a net-zero global integral of the surface heat flux (including shortwave radiation), would lead to

a non-zero integral due to a spatially varying factor $\alpha \rho^{-1}$, that for current planetary conditions leads to a net increase in sea

level. Similar conceptual processes occur for the mass flux, salt flux and geothermal flux.

**Appendix C: Specifying the impact of diffusive fluxes of salinity and temperature on density and sea level.**

Here an expression is derived for the impact of diffusive mixing on density and sea level. Mixing is represented in a mixing

tensor $\mathbf{K}$ ($\mathrm{m}^2\,\mathrm{s}^{-1}$) as a symmetric positive-definite kinematic diffusivity tensor that contains the contributions of the mesoscale

neutral and horizontal diffusion, and small-scale isotropic diffusion (Fox-Kemper et al., 2019), which can be written as

$$
\mathbf{K} = K_{\mathrm{N}} \left( \mathbf{I} - \hat{\mathbf{n}}_{\rho} \hat{\mathbf{n}}_{\rho} \right) + K_{\mathrm{H}} \left( \mathbf{I} - \hat{\mathbf{z}} \hat{\mathbf{z}} \right) + D\, \mathbf{k}\mathbf{k}.
\tag{C1}
$$

Here $\mathbf{I}$ is the identity tensor, The dia-neutral unit vector $\hat{\mathbf{n}}_{\rho} = \nabla \rho_l |\nabla \rho_l|^{-1}$ is defined according to the gradient of locally

referenced potential density $\rho_l$ (McDougall et al., 2014), where $\nabla \rho_l = \rho \left( -\alpha \nabla \Theta + \beta \nabla S \right)$. This allows us to write the mixing

tensor $\mathbf{K}$ as:

$$
\mathbf{K} \cdot \nabla \Theta = \underbrace{K_{\mathrm{H}} \nabla_{\mathrm{H}} \Theta}_{\mathrm{horizontal}} + \underbrace{K_{\mathrm{N}} \nabla_{\mathrm{N}} \Theta}_{\mathrm{neutral}} + \underbrace{D \frac{\partial \Theta}{\partial z} \mathbf{k}}_{\mathrm{vertical}},
\tag{C2}
$$

where $\nabla_{\mathrm{N}} \Theta = \left( \mathbf{I} - \hat{\mathbf{n}}_{\rho} \hat{\mathbf{n}}_{\rho} \right) \cdot \nabla \Theta$ and $\nabla_{\mathrm{H}} \Theta = \left( \mathbf{I} - \hat{\mathbf{z}} \hat{\mathbf{z}} \right) \cdot \nabla \Theta$. See McDougall et al. (2014) for a visual representation of the full

and small-slope rotation tensor. Combining Eqs. 1 with C1, the component of sea level rise that is only due to diffusive fluxes

of heat and salt that alter the density, can be written as (Griffies and Greatbatch, 2012; Groeskamp et al., 2019b):

$$
\left. \frac{\partial \eta}{\partial t} \right|_{\mathrm{diffusion}} = -\int_{-H}^{\eta} \nu \left( \alpha \nabla \cdot \mathbf{J}_{\mathrm{diff}}^{\Theta} - \beta \nabla \cdot \mathbf{J}_{\mathrm{diff}}^{S_{\mathrm{A}}} \right) dz.
\tag{C3}
$$

Here $\nu = \rho^{-1}$ is specific volume ($\mathrm{m}^3\,\mathrm{kg}^{-1}$). Writing the $\mathbf{J}_{\mathrm{diff}}$-terms as the density weighted down-gradient diffusive tracer

concentration fluxes for $S_{\mathrm{A}}$ and $\Theta$ gives:

$$
\mathbf{J}_{\mathrm{diff}}^{\Theta} = -\rho \mathbf{K} \cdot \nabla \Theta = \rho \mathbf{V}_{\Theta}, \quad \mathbf{J}_{\mathrm{diff}}^{S_{\mathrm{A}}} = -\rho \mathbf{K} \cdot \nabla S_{\mathrm{A}} = \rho \mathbf{V}_{S_{\mathrm{A}}}.
\tag{C4}
$$





It helps to define $\mathbf{V}_\Theta = -\mathbf{K} \cdot \nabla \Theta$ and $\mathbf{V}_{S_A} = -\mathbf{K} \cdot \nabla S_A$ as the down-gradient diffusive tracer flux of Conservative Temperature and Absolute Salinity, respectively. The minus sign in the expression for the $\mathbf{J}_{\mathrm{diff}}$-terms assure the down-gradient nature of the diffusive flux. Now using the following identities (Griffies and Greatbatch, 2012):

$$\nu\beta\nabla \cdot \rho\mathbf{V}_{S_A} \quad = \quad \nabla \cdot \beta\,\mathbf{V}_{S_A} - \mathbf{V}_{S_A} \cdot \nabla\beta - \beta\mathbf{V}_{S_A} \cdot \nabla\ln\rho, \tag{C5}$$

$$\nu\alpha\nabla \cdot \rho\mathbf{V}_\Theta \quad = \quad \nabla \cdot \alpha\,\mathbf{V}_\Theta - \mathbf{V}_\Theta \cdot \nabla\alpha - \alpha\mathbf{V}_\Theta \cdot \nabla\ln\rho, \tag{C6}$$

allows us to write

$$\nu\alpha\nabla \cdot \mathbf{J}_{\mathrm{diff}}^\Theta - \nu\beta\nabla \cdot \mathbf{J}_{\mathrm{diff}}^{S_A} \quad = \quad \nabla \cdot \mathbf{R} - \mathrm{P} - \mathbf{R} \cdot \nabla\ln\rho. \tag{C7}$$

With:

$$\mathbf{R} \quad = \quad -\alpha\mathbf{K} \cdot \nabla\Theta + \beta\mathbf{K} \cdot \nabla S_A = \alpha\mathbf{V}_\Theta - \beta\mathbf{V}_{S_A} = \mathbf{R}^{\mathrm{ntr}} + \mathbf{R}^{\mathrm{hor}} + \mathbf{R}^{\mathrm{ver}}, \tag{C8}$$

$$\mathrm{P} \quad = \quad \nabla\alpha \cdot (\mathbf{K} \cdot \nabla\Theta) - \nabla\beta \cdot (\mathbf{K} \cdot \nabla S_A) = \nabla\alpha \cdot \mathbf{V}_\Theta - \nabla\beta \cdot \mathbf{V}_{S_A}, \tag{C9}$$

$$\quad = \quad \mathrm{P}^{\mathrm{ntr}} + \mathrm{P}^{\mathrm{hor}} + \mathrm{P}^{\mathrm{ver}}.$$

Here $\mathbf{R}^{\mathrm{ntr}}$, $\mathbf{R}^{\mathrm{hor}}$, and $\mathbf{R}^{\mathrm{ver}}$ (m s$^{-1}$) are the components of $\mathbf{R}$ for the three different mixing direction, while $\mathrm{P}^{\mathrm{ntr}}$, $\mathrm{P}^{\mathrm{hor}}$, and $\mathrm{P}^{\mathrm{ver}}$ are the components of P for the three different mixing direction. The full expressions for $\mathbf{R}^{\mathrm{ntr}}$, $\mathbf{R}^{\mathrm{hor}}$, and $\mathbf{R}^{\mathrm{ver}}$ are given by

$$\mathbf{R}^{\mathrm{ntr}} \quad = \quad -K_\mathrm{N}\left[\alpha\,\nabla_\mathrm{N}\Theta - \beta\,\nabla_\mathrm{N}\,S_A\right] = 0,$$

$$\mathbf{R}^{\mathrm{hor}} \quad = \quad -K_\mathrm{H}\left[\alpha\,\nabla_\mathrm{H}\Theta - \beta\,\nabla_\mathrm{H}\,S_A\right],$$

$$\mathbf{R}^{\mathrm{ver}} \quad = \quad -D\left[-\alpha\,\frac{\partial\Theta}{\partial z} + \beta\,\frac{\partial S_A}{\partial z}\right]\mathbf{k}. \tag{C10}$$

By definition, for the neutral direction $\alpha\nabla_\mathrm{N}\Theta = \beta\nabla_\mathrm{N}S_A$ and therefore $\mathbf{R}^{\mathrm{ntr}} = 0$. The full expressions for $\mathrm{P}^{\mathrm{ntr}}$, $\mathrm{P}^{\mathrm{hor}}$, and $\mathrm{P}^{\mathrm{ver}}$ are given by

$$\mathrm{P}^{\mathrm{ntr}} \quad = \quad K_\mathrm{N}\left(\nabla_\mathrm{N}\alpha \cdot \nabla_\mathrm{N}\Theta - \nabla_\mathrm{N}\beta \cdot \nabla_\mathrm{N}S_A\right) = \mathrm{P}_{\mathrm{T_b}}^{\mathrm{ntr}} + \mathrm{P}_{\mathrm{C_b}}^{\mathrm{ntr}},$$

$$\mathrm{P}^{\mathrm{hor}} \quad = \quad K_\mathrm{H}\left(\nabla_\mathrm{H}\alpha \cdot \nabla_\mathrm{H}\Theta - \nabla_\mathrm{H}\beta \cdot \nabla_\mathrm{H}S_A\right) = \mathrm{P}_{\mathrm{T_b}}^{\mathrm{hor}} + \mathrm{P}_{\mathrm{C_b}}^{\mathrm{hor}},$$

$$\mathrm{P}^{\mathrm{ver}} \quad = \quad D\left(\frac{\partial\alpha}{\partial z}\,\frac{\partial\Theta}{\partial z} - \frac{\partial\beta}{\partial z}\,\frac{\partial S_A}{\partial z}\right) = \mathrm{P}_{\mathrm{T_b}}^{\mathrm{ver}} + \mathrm{P}_{\mathrm{C_b}}^{\mathrm{ver}}. \tag{C11}$$

Where the production terms in Eq. C11 are further expanded into the more well know cabbeling and thermobaricity components, for which the expression are provided in Eq. C17 to Eq. C22 in Appendix C1. When inserting Eq. (C7) into Eq. (C3) and applying the Leibniz integral rule for differentiation under an integral to rewrite the $\nabla \cdot \mathbf{R}$ term (see Appendix C2 ), to obtain

$$\left.\frac{\partial\eta}{\partial t}\right|_{\mathrm{diffusion}} \quad = \quad -\nabla_\mathrm{H} \cdot \int_{-H}^\eta \mathbf{R}\,dz + \mathbf{R}(\eta) \cdot \nabla_\mathrm{H}\eta - \mathbf{R}(-H) \cdot \nabla_\mathrm{H}(-H)$$

$$+ \int_{-H}^\eta \mathrm{P}\,dz + \int_{-H}^\eta \mathbf{R} \cdot \nabla ln(\rho)\,dz. \tag{C12}$$



As there are no diffusive fluxes through any of the ocean boundaries, a global integral of $\left.\frac{\partial \eta}{\partial t}\right|_{\text{diffusion}}$ would cause the first term on the right hand side to vanish. Hence, comparable to volume redistribution by ocean currents, this term locally changes sea

level without a net global effect. Even though this only applies to the first term on the r.h.s. of Eq. C12 where $\mathbf{R}$ is involved, this inspired the naming of "$\mathbf{R}$" as "redistribution" term. All other terms in the equation, have both a local and net global contribution to GMSL. Of special interest is the term P ($\text{s}^{-1}$), that is directly related to cabbeling and thermobaricity in all three mixing direction (McDougall, 1987b), as detailed in Appendix C. To further develop the impact of ocean mixing on sea level rise, the following steps are applied. First 1) use that there are no fluxes through the boundaries, thus $\mathbf{R}^{\text{ver}}(H) = \mathbf{R}^{\text{ver}}(\eta) = 0$, and the

vertical integral of $\partial \mathbf{R}^{\text{ver}}/\partial z$ is zero, 2) write $P = P_{\text{T}_{\flat}}^{\text{ntr}} + P_{\text{C}_{\flat}}^{\text{ntr}} + P_{\text{C}_{\flat}}^{\text{hor}} + P_{\text{T}_{\flat}}^{\text{hor}} + P_{\text{T}_{\flat}}^{\text{ver}} + P_{\text{C}_{\flat}}^{\text{ver}}$, and 3) rewrite $\nabla \ln \rho \cdot \mathbf{R}$ using the identity $\nabla ln(\rho) \cdot \mathbf{R} = \nu \mathbf{R} \cdot \nabla \rho$ in combination with the specific mixing direction to write these terms from Eq. C12 as:

$$\nabla_{\text{H}} ln(\rho) \cdot \mathbf{R}^{\text{hor}} = K_{\text{H}}^{-1} \left|\mathbf{R}^{\text{hor}}\right|^2 + \kappa \nabla_{\text{H}} P \cdot \mathbf{R}^{\text{hor}} \approx K_{\text{H}}^{-1} \left|\mathbf{R}^{\text{hor}}\right|^2 \tag{C13}$$

$$\frac{\partial ln(\rho)}{\partial z} \mathbf{k} \cdot \mathbf{R}^{\text{ver}} = D^{-1} \left|\mathbf{R}^{\text{ver}}\right|^2 - \rho g \kappa \mathbf{k} \cdot \mathbf{R}^{\text{ver}} \tag{C14}$$

$$\nabla_{\text{N}} ln(\rho) \cdot \mathbf{R}^{\text{ntr}} = K_{\text{N}}^{-1} \left|\mathbf{R}^{\text{ntr}}\right|^2 + \kappa \nabla_{\text{N}} P \cdot \mathbf{R}^{\text{ntr}} \tag{C15}$$

Inserting this all these points into Eq. C12, leaves the final expression for the impact of diffusive fluxes on sea level:

$$
\begin{aligned}
\left.\frac{\partial \eta}{\partial t}\right|_{\text{diffusion}} \approx \quad & \underbrace{-\nabla_{\text{H}} \cdot \int_{-H}^{\eta} \mathbf{R}^{\text{hor}}\, dz + \mathbf{R}^{\text{hor}}(\eta) \cdot \nabla_{\text{H}} \eta - \mathbf{R}^{\text{hor}}(-H) \cdot \nabla_{\text{H}}(-H)}_{\text{Redistribution}} \\[2mm]
& + \int_{-H}^{\eta} K_{\text{H}}^{-1} \left|\mathbf{R}^{\text{hor}}\right|^2\, dz + \int_{-H}^{\eta} D^{-1} \left|\mathbf{R}^{\text{ver}}\right|^2\, dz - \int_{-H}^{\eta} \rho g \kappa \mathbf{k} \cdot \mathbf{R}^{\text{ver}}\, dz \\[2mm]
& + \int_{-H}^{\eta} P_{\text{T}_{\flat}}^{\text{ntr}} + P_{\text{C}_{\flat}}^{\text{ntr}} + P_{\text{C}_{\flat}}^{\text{hor}} + P_{\text{T}_{\flat}}^{\text{ver}} + P_{\text{C}_{\flat}}^{\text{ver}}\, dz. \tag{C16}
\end{aligned}
$$

## C1  The Production terms expanded

The production term of Eq. (C10) can be rewritten using the mixing tensor of Eq. (C2) into the more well know cabbeling and thermobaricity components. The expression below allow us to see the similarities between thermobaricity and cabbeling





(densification upon mixing) for the different mixing direction:

$$P_{T_b}^{ntr} = K_N\, T_b\, \nabla_N P \cdot \nabla_N \Theta, \tag{C17}$$

$$P_{C_b}^{ntr} = K_N\, C_b\, |\nabla_N \Theta|^2, \tag{C18}$$

$$P_{T_b}^{hor} = K_H\, \nabla_H P \cdot \left( \frac{\partial \alpha}{\partial p}\, \nabla_H \Theta - \frac{\partial \beta}{\partial p}\, \nabla_H S_A \right)$$

$$= K_H\, T_b^{hor}\, \nabla_H P \cdot \nabla_H \Theta \tag{C19}$$

$$P_{C_b}^{hor} = K_H\, \left( \frac{\partial \alpha}{\partial \Theta}\, |\nabla_H \Theta|^2 + 2\, \frac{\partial \alpha}{\partial S_A}\, \nabla_H S_A \cdot \nabla_H \Theta - \frac{\partial \beta}{\partial S_A}\, |\nabla_H S_A|^2 \right)$$

$$= K_H\, C_b^{hor}\, |\nabla_H \Theta|^2 \tag{C20}$$

$$P_{T_b}^{ver} = -D\, \rho\, g\, \left( \frac{\partial \alpha}{\partial P}\, \frac{\partial \Theta}{\partial z} - \frac{\partial \beta}{\partial P}\, \frac{\partial S_A}{\partial z} \right)$$

$$= -D\, \rho\, g\, T_b^{ver}\, \frac{\partial \Theta}{\partial z} \tag{C21}$$

$$P_{C_b}^{ver} = D\, \left( \frac{\partial \alpha}{\partial \Theta}\, \left|\frac{\partial \Theta}{\partial z}\right|^2 + 2\, \frac{\partial \alpha}{\partial S_A}\, \frac{\partial S_A}{\partial z}\, \frac{\partial \Theta}{\partial z} - \frac{\partial \beta}{\partial S_A}\, \left|\frac{\partial S_A}{\partial z}\right|^2 \right)$$

$$= D\, C_b^{ver}\, \left|\frac{\partial \Theta}{\partial z}\right|^2 \tag{C22}$$

In order to break down Eq. C11 into Eqs. (C17) - (C22), the following identities are used, before a number of new variables are defined. First of all, $C_b$ (K$^{-2}$) and $T_b$ (Pa$^{-1}$ K$^{-1}$) are the cabbeling and thermobaricity coefficients for the neutral direction,
as previously defined by (McDougall, 1984, 1987a) given by

$$C_b = \frac{\partial \alpha}{\partial \Theta} + 2\, \frac{\alpha}{\beta}\, \frac{\partial \alpha}{\partial S_A} - \frac{\alpha^2}{\beta^2}\, \frac{\partial \beta}{\partial S_A}, \quad T_b = \frac{\partial \alpha}{\partial P} - \frac{\alpha}{\beta}\, \frac{\partial \beta}{\partial P}, \tag{C23}$$

In a similar fashion I here introduce $C_b^{ver}$ and $C_b^{hor}$ (K$^{-2}$) and $T_b^{ver}$ and $T_b^{hor}$ (Pa$^{-1}$ K$^{-1}$) as the vertical and horizontal equivalents of their neutral counterparts, now given by:

$$C_b^{hor} = \frac{\partial \alpha}{\partial \Theta} + 2\, \frac{\alpha}{\beta}\, \frac{\partial \alpha}{\partial S_A}\, \frac{1}{R_H} - \frac{\alpha^2}{\beta^2}\, \frac{\partial \beta}{\partial S_A}\, \frac{1}{R_H^2}, \quad T_b^{hor} = \frac{\partial \alpha}{\partial P} - \frac{\alpha}{\beta}\, \frac{\partial \beta}{\partial P}\, \frac{1}{R_H}, \tag{C24}$$

$$C_b^{ver} = \frac{\partial \alpha}{\partial \Theta} + 2\, \frac{\alpha}{\beta}\, \frac{\partial \alpha}{\partial S_A}\, \frac{1}{R_\rho} - \frac{\alpha^2}{\beta^2}\, \frac{\partial \beta}{\partial S_A}\, \frac{1}{R_\rho^2}, \quad T_b^{ver} = \frac{\partial \alpha}{\partial P} - \frac{\alpha}{\beta}\, \frac{\partial \beta}{\partial P}\, \frac{1}{R_\rho}. \tag{C25}$$

Here

$$R_\rho = \frac{\alpha}{\beta}\, \frac{\frac{\partial \Theta}{\partial z}}{\frac{\partial S_A}{\partial z}}, \quad \text{and} \quad R_H = \frac{\alpha}{\beta}\, \frac{\nabla_H \Theta}{\nabla_H S_A} \tag{C26}$$

are the vertical $R_\rho$ and horizontal stability ratio $R_H$. Although the latter may have less physical meaning in turbulence theory, it is of symbolic use for comparing between the newly defined horizontal and vertical cabbeling and thermobaricity terms. To
obtain Eq. (C23-C25), use that $\rho$, $\alpha$, $\beta$ and $\kappa$ are given by polynomials, such that Clairaut's theorem can be used:

$$\frac{\partial}{\partial \varphi}\, \frac{1}{\rho}\, \frac{\partial \rho}{\partial \phi} = \frac{1}{\rho}\, \frac{\partial^2 \rho}{\partial \varphi \partial \phi} + \frac{1}{\rho^2}\, \frac{\partial \rho}{\partial \phi}\, \frac{\partial \rho}{\partial \varphi} = \frac{1}{\rho}\, \frac{\partial^2 \rho}{\partial \phi \partial \varphi} + \frac{1}{\rho^2}\, \frac{\partial \rho}{\partial \varphi}\, \frac{\partial \rho}{\partial \phi} = \frac{\partial}{\partial \phi}\, \frac{1}{\rho}\, \frac{\partial \rho}{\partial \varphi}. \tag{C27}$$





This allows us to fill in the nine different combination (using $S_A$, $\Theta$ or $P$) and obtain the following identities (McDougall, 1984, 1987a):

$$-\frac{\partial\alpha}{\partial S_A} = \frac{\partial\beta}{\partial\Theta}, \quad -\frac{\partial\alpha}{\partial P} = \frac{\partial\kappa}{\partial\Theta}, \quad \frac{\partial\beta}{\partial P} = \frac{\partial\kappa}{\partial S_A}. \tag{C28}$$

In addition use that:

$$\nabla\alpha = \frac{\partial\alpha}{\partial\Theta}\nabla\Theta + \frac{\partial\alpha}{\partial S_A}\nabla S_A + \frac{\partial\alpha}{\partial p}\nabla P, \tag{C29}$$

$$\nabla\beta = \frac{\partial\beta}{\partial\Theta}\nabla\Theta + \frac{\partial\beta}{\partial S_A}\nabla S_A + \frac{\partial\beta}{\partial p}\nabla P, \tag{C30}$$

**C2    Leibniz Rule applied to R**

To obtain Eq.(C12) from Eq. (C3) the Leibniz integral rule for differentiation under an integral is used:

$$\frac{\partial}{\partial x}\int_{a(x)}^{b(x)} f(x,t)dt = \int_{a(x)}^{b(x)}\frac{\partial f(x,t)}{\partial x}dt + f(x,b(x))\frac{\partial b(x)}{\partial x} - f(x,a(x))\frac{\partial a(x)}{\partial x} \tag{C31}$$

This can also be written as:

$$\int_{a(x)}^{b(x)}\frac{\partial f(x,t)}{\partial x}dt = \frac{\partial}{\partial x}\int_{a(x)}^{b(x)} f(x,t)dt - f(x,b(x))\frac{\partial b(x)}{\partial x} + f(x,a(x))\frac{\partial a(x)}{\partial x} \tag{C32}$$

Using $\mathbf{R} = (R_x, R_y, R_z)$, this allows us to write:

$$\int_{-H}^{\eta}\frac{\partial R_x}{\partial x}dz = \frac{\partial}{\partial x}\int_{-H}^{\eta} R_x dz - R_x(\eta)\frac{\partial\eta}{\partial x} + R_x(-H)\frac{\partial(-H)}{\partial x} \tag{C33}$$

$$\int_{-H}^{\eta}\frac{\partial R_y}{\partial y}dz = \frac{\partial}{\partial y}\int_{-H}^{\eta} R_y dz - R_y(\eta)\frac{\partial\eta}{\partial y} + R_y(-H)\frac{\partial(-H)}{\partial y} \tag{C34}$$

$$\int_{-H}^{\eta}\frac{\partial R_z}{\partial z}dz = R_z(\eta) - R_z(-H) = 0 \quad \text{No diffusion through boundary} \tag{C35}$$

Which leaves:

$$\int_{-H}^{\eta}\nabla\cdot\mathbf{R}dz = \nabla_H\cdot\int_{-H}^{\eta}\mathbf{R}\,dz - \mathbf{R}(\eta)\cdot\nabla_H\eta + \mathbf{R}(-H)\cdot\nabla_H(-H) \tag{C36}$$

**Appendix D:  Specifying the sea level change due to eddy-induced transport**

Here the impact of quasi-Stokes transport on sea level is derived. First the eddy-induced velocity $(u^*, v^*, w^*)$ is defined using the eddy induced transport $\mathbf{\Upsilon}$ as:

$$\mathbf{u}_{\text{eddy}} = (u^*, v^*, w^*) = \left(\frac{\partial\mathbf{\Upsilon}}{\partial z}, -\nabla\cdot\mathbf{\Upsilon}\right). \tag{D1}$$





Ferrari et al. (2010) argued that the eddy induced transport should be zero at the ocean surface and bottom $\mathbf{\Upsilon}(\eta) = \mathbf{\Upsilon}(-H) = 0$, to ensure a zero barotropic component due to the eddy-induced velocity. This means that the vertical integral of the eddy-induced velocity, which is a component of $\mathbf{u}$ in Eq. 1, is zero. Yet, the eddy-induced transport will impact density by transporting salt and heat. Following Griffies (1998), this impact can be written as a density weighted skew flux given by:

$$\mathbf{J}_{\text{stir}}^{\Theta} = -\rho\mathbf{K}_{\text{stir}} \cdot \nabla\Theta = \rho\mathbf{V}_{\text{stir}}^{\Theta}, \quad \mathbf{J}_{\text{stir}}^{S_{\text{A}}} = -\rho\mathbf{K}_{\text{stir}} \cdot \nabla S_{\text{A}} = \rho\mathbf{V}_{\text{stir}}^{S_{\text{A}}}. \tag{D2}$$

Where:

$$\mathbf{K}_{\text{stir}} = \begin{pmatrix} 0 & 0 & \Upsilon^{(x)} \\ 0 & 0 & \Upsilon^{(y)} \\ -\Upsilon^{(x)} & -\Upsilon^{(y)} & 0 \end{pmatrix}. \tag{D3}$$

and

$$\mathbf{V}_{\text{stir}}^{\Theta} = -\mathbf{K}_{\text{stir}} \cdot \nabla\Theta = \left(\Upsilon^x\frac{\partial\Theta}{\partial z}, \Upsilon^y\frac{\partial\Theta}{\partial z}, -\mathbf{\Upsilon} \cdot \nabla_H\Theta\right) \tag{D4}$$

$$\mathbf{V}_{\text{stir}}^{S_{\text{A}}} = -\mathbf{K}_{\text{stir}} \cdot \nabla S_{\text{A}} = \left(\Upsilon^x\frac{\partial S_{\text{A}}}{\partial z}, \Upsilon^y\frac{\partial S_{\text{A}}}{\partial z}, -\mathbf{\Upsilon} \cdot \nabla_H S_{\text{A}}\right)$$

Hence $\mathbf{V}_{\text{stir}}^{\Theta}$ and $\mathbf{V}_{\text{stir}}^{S_{\text{A}}}$ are the down gradient eddy tracer flux of temperature and salinity, respectively. Following Gent et al. (1995); McDougall and McIntosh (2001); Ferrari et al. (2010) the eddy transport flux (m$^2$ s$^{-1}$) is defined as:

$$\Upsilon^x = -K_{\text{stir}} S_x f_{\text{stir}}(z), \quad \Upsilon^y = -K_{\text{stir}} S_y f_{\text{stir}}(z), \tag{D5}$$

Here $K_{\text{stir}}$ is the stirring strength, also know as the "GM diffusivity" (Gent and McWilliams, 1990), $S_x$ and $S_y$ are neutral slopes and $f_{\text{stir}}(z)$ is some vertical tapering function that assures that $\mathbf{\Upsilon} = (\Upsilon^x, \Upsilon^y)$ satisfies the boundary condition $\mathbf{\Upsilon}(\eta) = \mathbf{\Upsilon}(-H) = 0$. for the vertical tapering function $f_{\text{stir}}(z)$ a linear tapering between 0 and 1 is used, over the upper 400m from the surface down and from the bottom up. Analogue to Eq. C7 it is found that:

$$\nu\alpha\nabla \cdot \mathbf{J}_{\text{stir}}^{\Theta} - \nu\beta\nabla \cdot \mathbf{J}_{\text{stir}}^{S_{\text{A}}} = \nabla \cdot \mathbf{R}_{\text{stir}} - \text{P}_{\text{stir}} - \mathbf{R}_{\text{stir}} \cdot \nabla\ln\rho. \tag{D6}$$

With:

$$\begin{aligned}
\mathbf{R}_{\text{stir}} &= \alpha\mathbf{V}_{\text{stir}}^{\Theta} - \beta\mathbf{V}_{\text{stir}}^{S_{\text{A}}}, \\
&= \Upsilon^x\left[\alpha\frac{\partial\Theta}{\partial z} - \beta\frac{\partial S_{\text{A}}}{\partial z}\right]\mathbf{i} + \Upsilon^y\left[\alpha\frac{\partial\Theta}{\partial z} - \beta\frac{\partial S_{\text{A}}}{\partial z}\right]\mathbf{j} \\
&\quad - \Upsilon^x\left[\alpha\frac{\partial\Theta}{\partial x} - \beta\frac{\partial S_{\text{A}}}{\partial x}\right]\mathbf{k} - \Upsilon^y\left[\alpha\frac{\partial\Theta}{\partial y} - \beta\frac{\partial S_{\text{A}}}{\partial y}\right]\mathbf{k},
\end{aligned} \tag{D7}$$

$$\begin{aligned}
\text{P}_{\text{stir}} &= \nabla\alpha \cdot \mathbf{V}_{\text{stir}}^{\Theta} - \nabla\beta \cdot \mathbf{V}_{\text{stir}}^{S_{\text{A}}} \\
&= \Upsilon^x\left(\frac{\partial\Theta}{\partial z}\frac{\partial\alpha}{\partial x} - \frac{\partial\Theta}{\partial x}\frac{\partial\alpha}{\partial z} - \frac{\partial S_{\text{A}}}{\partial z}\frac{\partial\beta}{\partial x} + \frac{\partial S_{\text{A}}}{\partial x}\frac{\partial\beta}{\partial z}\right) \\
&\quad + \Upsilon^y\left(\frac{\partial\Theta}{\partial z}\frac{\partial\alpha}{\partial y} - \frac{\partial\Theta}{\partial y}\frac{\partial\alpha}{\partial z} - \frac{\partial S_{\text{A}}}{\partial z}\frac{\partial\beta}{\partial y} + \frac{\partial S_{\text{A}}}{\partial y}\frac{\partial\beta}{\partial z}\right).
\end{aligned} \tag{D8}$$





Inserting Eq. D6 into Eq. 1, and using the Leibniz rule again, gives the impact of stirring on sea level evolution given by:

$$
\quad \left.\frac{\partial \eta}{\partial t}\right|_{\text{stirring}} = -\int_{-H}^{\eta} \nu \left(\alpha \nabla \cdot \mathbf{J}_{\text{stir}}^{\Theta} - \beta \nabla \cdot \mathbf{J}_{\text{stir}}^{S_A}\right) dz
$$

$$
= -\nabla_{\text{H}} \cdot \int_{-H}^{\eta} \mathbf{R}_{\text{stir}} \, dz + \int_{-H}^{\eta} \mathrm{P}_{\text{stir}} \, dz + \int_{-H}^{\eta} \mathbf{R}_{\text{stir}} \cdot \nabla \ln \rho \, dz. \tag{D9}
$$

Here it is used that the boundaries $\mathbf{\Upsilon} = 0$ and that the vertical integral of $\mathbf{R}_{\text{stir}}$ is zero. Now using the same identity as for Eq. C13 - Eq. C15, for coding purposes, the the last term on the r.h.s. is expanded as:

$$
\int_{-H}^{\eta} \mathbf{R}_{\text{stir}} \cdot \nabla \ln \rho \, dz = \int_{-H}^{\eta} \nu \mathbf{R}_{\text{stir}} \cdot \nabla \rho \, dz
$$

$$
\qquad\qquad = \int_{-H}^{\eta} \mathbf{R}_{\text{stir}} \cdot \left(-\alpha \nabla \Theta + \beta \nabla S_A + \kappa \nabla P\right) dz
$$

$$
= \int_{-H}^{\eta} \Upsilon^x \left[\alpha \frac{\partial \Theta}{\partial z} - \beta \frac{\partial S_A}{\partial z}\right] \left(-\alpha \frac{\partial \Theta}{\partial x} + \beta \frac{\partial S_A}{\partial x} + \kappa \frac{\partial P}{\partial x}\right) dz
$$

$$
+ \int_{-H}^{\eta} \Upsilon^y \left[\alpha \frac{\partial \Theta}{\partial z} - \beta \frac{\partial S_A}{\partial z}\right] \left(-\alpha \frac{\partial \Theta}{\partial y} + \beta \frac{\partial S_A}{\partial y} + \kappa \frac{\partial P}{\partial y}\right) dz
$$

$$
+ \int_{-H}^{\eta} -\left(\Upsilon^x \left[\alpha \frac{\partial \Theta}{\partial x} - \beta \frac{\partial S_A}{\partial x}\right] + \Upsilon^y \left[\alpha \frac{\partial \Theta}{\partial y} - \beta \frac{\partial S_A}{\partial y}\right]\right) \left(-\alpha \frac{\partial \Theta}{\partial z} + \beta \frac{\partial S_A}{\partial z} - \kappa \rho g\right) dz
$$

$$
\tag{D10}
$$

Through Eq. D5, stirring depends on the accuracy of the calculated neutral slope. This allows us to define the impact of non-neutrality on stirring and sea level as:

$$
\Upsilon^x_{\text{non−neutral}} = -K_{\text{stir}} \left(S_x - S_x^{(\text{perfectly neutral})}\right) f_{\text{stir}}(z),
$$

$$
\Upsilon^y_{\text{non−neutral}} = -K_{\text{stir}} \left(S_y - S_y^{(\text{perfectly neutral})}\right) f_{\text{stir}}(z). \tag{D11}
$$

An overestimate of the neutral slopes will lead to more reduction in GMSL by eddy stirring.

**Appendix E: Balancing the heat fluxes**

To construct the balanced heat and mass fluxes, I want to distribute the global mass or heat imbalance over all grid points and time, proportionally to the local flux. Hence the larger the flux terms, the larger the fraction of the imbalance this grid point will compensate for.

Here the mass flux is used to illustrate the procedure, but the same procedure is applied to the heat fluxes. First the global 750 imbalances $\epsilon$ (kg s$^{-1}$) is determined for the different mass flux components (note that the contribution for evaporation $E$ are





split in a positive and negative part):

$$\epsilon_P \;=\; \int\limits_{\text{global}} P\,dA, \quad \epsilon_R = \int\limits_{\text{global}} R\,dA, \tag{E1}$$

$$\epsilon_{E>0} \;=\; \int\limits_{\text{global}} (E>0)\,dA, \quad \epsilon_{E<0} = \int\limits_{\text{global}} (E<0)\,dA \tag{E2}$$

Then calculate the total net flux imbalance $\epsilon_q$ as well as the total flux exchange that has occurred $\epsilon_{|q|}$, which is the sum of the absolute values:

$$\epsilon_q \;=\; \epsilon_P + \epsilon_R + \epsilon_{E<0} + \epsilon_{E>0}, \tag{E3}$$

$$\epsilon_{|q|} \;=\; |\epsilon_P| + |\epsilon_R| + |\epsilon_{E<0}| + |\epsilon_{E>0}| + |\epsilon_{E>0}|. \tag{E4}$$

Note that $\epsilon_q$ is equal to $\int_{\text{global}} Q_{\mathrm{m}}\,dA$ and thus the total global net mass flux. Now define a vector of the individual flux terms

and exchange terms:

$$\mathbf{L} \;=\; [\epsilon_P, \epsilon_R, \epsilon_{E<0}, \epsilon_{E>0}], \tag{E5}$$

$$\mathbf{L}_{\mathrm{A}} \;=\; [\,|\epsilon_P|, |\epsilon_R|, |\epsilon_{E<0}|, |\epsilon_{E>0}|\,]. \tag{E6}$$

The above definition are used to redistribute the imbalance over all the terms. Then compute the fraction $r$ that each term should compensate for, with respect to the total exchange with the ocean:

$$r \;=\; \epsilon_{|q|}^{-1}\,\mathbf{L}_{\mathrm{A}} \tag{E7}$$

The sum of $r$ is 1, and each number is the fraction that a mass flux term needs to compensate in total. Using this, the new imbalances are defined (e.g. $\epsilon_P^0$ instead of $\epsilon_P$) that are the new globally integrated total fluxes for each individual term, such

that the net global imbalance adds up to zero:

$$\left[\epsilon_P^0, \epsilon_R^0, \epsilon_{E<0}^0, \epsilon_{E>0}^0\right] \;=\; \mathbf{L} - r\,\epsilon_q = \mathbf{L} - \frac{\epsilon_q}{\epsilon_{|q|}}\,\mathbf{L}_{\mathrm{A}} \tag{E8}$$

The new "balanced" mass flux then becomes:

$$P_{\mathrm{b}}(x,y,t) \;=\; P(x,y,t)\frac{\epsilon_P^0}{\epsilon_P} \tag{E9}$$

$$R_{\mathrm{b}}(x,y,t) \;=\; R(x,y,t)\frac{\epsilon_R^0}{\epsilon_R} \tag{E10}$$

$$(E>0)_{\mathrm{b}}(x,y,t) \;=\; (E>0)(x,y,t)\frac{\epsilon_{E>0}^0}{\epsilon_{E>0}} \tag{E11}$$

$$(E<0)_{\mathrm{b}}(x,y,t) \;=\; (E<0)(x,y,t)\frac{\epsilon_{E<0}^0}{\epsilon_{E<0}} \tag{E12}$$

$$E_{\mathrm{b}}(x,y,t) \;=\; (E>0)_{\mathrm{b}}(x,y,t) + (E<0)_{\mathrm{b}}(x,y,t) \tag{E13}$$

A similar exercise is done with the heat fluxes.





*Author contributions.* Sjoerd Groeskamp is the sole author.

*Competing interests.* The author is a member of the editorial board of Ocean Science (OS)

*Acknowledgements.* I thank Stephen Griffies for useful comments on an early draft of this work, Anja Mödl for doing some early calculations on this work with me, as part of her MSc thesis. I thank Ryan Abernathey, Magnus Hieronymus for discussions on some of the nonlinear equation of state stuff in this paper. I thank Aimee Slangen for useful discussion on this topic.



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
