# Peer review of "Observational-based quantification of physical processes that impact the evolution of global mean sea level"

_EGUsphere, 2025_

## Author Comment (AC1)

**Observational-based quantification of physical processes that impact the evolution of global mean sea level**

Sjoerd Groeskamp

**Response to Reviewer #1**

The basic premise of this manuscript is built around a distinction between a top-down and bottom-up approach to estimating sea level rise. This manuscript approaches the problem from the bottom up, examining the imbalance in mass and heat inputs to infer net sea level rise. One might alternatively refer to this, as a flux approach rather than a reservoir approach. The flux approach is exceedingly difficult since it requires accurate measurements of rainfall, evaporation, and storm-related heat gain and losses, all of which depend on transient or extreme events. Thus, as the author acknowledges, there is low probability of obtaining a closed budget through this approach. That could make this study seem like an unusual intellectual exercise, but that view would undersell the value of the manuscript. In fact the author avoids the global imbalance problem by working with fluxes that have been adjusted to remove long-term drift, using the fields to assess the impacts of mixing and advection, which can modify sea level, particularly through effects of cabbeling and thermobaricity. The exercise is valuable as a strategy for exploring the mechanisms underpinning sea level rise and for evaluating terms that need to be handled correctly in modeling studies. The maps in the manuscript provide a useful assessment of the relative importance of the less intuitive processes that contribute to the global sea level rise budget. In this sense, the manuscript is a valuable and thorough contribution to discussions of global sea level rise.

Thanks for this summary and analyses. I'm glad to see that the idea came across and I'm encouraged by the positive comments. I want to state that I'm impressed and grateful for the reviewer's effort to read and improve this paper. I apologize for the spelling mistakes.

The analysis framework appears sound. (I admit that I didn't check every step of the derivation.) The results provide a useful point of reference for evaluating contributors for sea level rise, and I think the results will be useful to journal readers. There are a number of details that should be addressed prior to publication.

1. The reference list is thorough but omits some recent studies that have been important to the global sea level rise discussion (e.g. Hamlington et al, Nature Communications Earth and Environment, 2024; Fasullo and Nerem, PNAS, 2018; Nerem et al, PNAS, 2018; Nerem et al, Earth's Future, 2022). It's not clear that all of these are relevant, but to my mind this body of work is relevant for framing what the "top down" approach for analyzing sea level rise.

Great! I found a place for all these references and more in the first 2 paragraphs of the introduction. Note the first paragraphs of the introduction are thoroughly rewritten.

2. Line 100. Equation (4) is described as deriving from equations (1) and (3). A direct substitution of (3) into (1) will not yield (4), so a bit of additional explanation is needed.

Ah yes, this was because I was only considering the first term on the right-hand side of equation 1. Then it does follows directly. So, I clarified this now to avoid confusion (L107). Thanks.

3. Line 196. "It also showcases". It's not clear that Equation (15) shows that sea level is constantly evolving---it shows that many terms contribute to sea level evolution. Perhaps there's a better way to describe the fact that sea level is the result of many contributing factors.

Yes, I see your point. I have adapted the sentence and removed a part of it. See L205-L208.

4. Line 410. "densification upon mixing will keep the ocean from ever expanding". This statement seems likely to be too strong. I think the intent is to say that densification upon mixing will counteract expansion due to nonlinear thermal effects.

It was meant as a more "romantic" way of saying the same thing. But this is not a novel, so I have now changed the sentence to be more scientific. Yet, part of it is still there. This is because somehow attention needs to be pointed towards the fact that this is not with respect to "normal expansion". This expansion always happens, even when there is no net warming of the ocean. It is hard to get this message across, without first going into detail. I was hoping this way could be a good start, before the details come. I hope you consider the current way it is embedded in the text as an improvement.

5. Figures 1-8. The figures in this manuscript show global maps plotted over what appears to be the same domain. However, the aspect ratios of the figures vary substantially, in some cases even within the same figure (e.g. Figures 3 and 4). This makes it difficult for readers to compare the images. The maps should be redone with a single, consistent projection, even if that requires a white gap within the figure. In addition, colorbars are presented with inconsistent labeling (e.g. Figure 6)

Thanks for this comment. I have played around with the figures, colorbars, titles and aspect ratios. I think I have found a way that they all have the same aspect ratio throughout the whole manuscript. With all the other rearrangements as well, I think this has improved the presentation and paper.

6. Line 551-553. "Nonlinear thermal expansion and densification upon mixing go hand in hand, are of the same order of magnitude and of opposite sign, therewith compensating each other." The manuscript earlier explains that thermobaricity is about an order of magnitude smaller than mixing-related densification. This point needs to be clarified so that readers can connect this sentence with material earlier in the paper to understand which terms correspond to thermal expansion and why this differs from the earlier thermobaricity/densification discussion where the terms are not the same order of magnitude.

Nonlinear thermal expansion and thermobaricity are not the same thing. Thermobaricity is related to the expansion coefficient being a function of pressure, while the nonlinear thermal expansion is related to the expansion coefficient being a function of, mainly temperature. Thermobaricity is an order of magnitude smaller than cabbeling. Nonlinear thermal expansion is of the same order as cabbeling. To address this, I made the following clarifications:

- L65, introduction.
- L157, section 2.3 clarifies the meaning of thermobaricity. But not in relation to nonlinear thermal expansion.
- 7. The appendices are intentionally derivation heavy. This makes them difficult to verify. Notably, the brevity of Appendix C is good, but makes it hard to unravel. It would be useful to explain the difference between z-hat and k. Both are presumably unit vectors. Are they in the same direction? The author should make sure that terms are clearly defined

Thank you for picking this up! This was indeed unnecessary clutter. I have gone through the appendix once again and tried to clarify where I could.

8. Minor points of grammar. The manuscript contains lots of typos, missing words, etc, and a revision should be carefully proofread. I have indicated issues that I noted, but I'm sure that there are more:

Wow, what an effort! And to think that, when I submitted it, I thought I got most mistakes. I thank the reviewer a lot for the effort to point at all these grammar problems. I fixed them all. I have really done my best, not to leave many new ones after the revisions.

```
Line 46: ", which raises" --> "and raise"
Line 46: "Result" --> "Results"
Line 68: "presented and derived" --> "derived" (More compact wording seems sufficient.)
Line 71: "amongs" --> "among"
Line 72: add comma after "(section 5)"
```

Line 88: "coined" --> "identified as" or omit "coined"

Line 88: "of Eq. 1" --> "in Eq. 1"

Line 84: "is studied" --> "are studied"

Line 98. No verb. Change equation (3) punctuation to be a comma, and change "Where" to "where"

Line 102. "conceptual" --> "conceptually"

Line 109. "therewith density. Mainly" --> "thus density, mainly".

Line 111. Missing period. "sensible heat fluxes" --> "sensible heat fluxes."

Line 118. "are due" v "is due"

Line 119. What does "O(meter)" refer to? Should this be "meter-scale eddies", or is there a different intended meaning?

Line 122. "eddies O(20-200 km)" --> "eddies of O(20-200 km)"

```
Line 140. "also creates" --> "we also create"
Line 141. "be interpret as interaction" --> "be interpreted as the interaction"
Line 142. "advantages" --> "advantageous"
Line 143. "of redistribution" --> "of the redistribution"
Line 146. "impact is" --> "impact"
Line 149. "naming" --> "terminology"
Line 150. "direction" --> "directions"
Line 161. "non-resolved" --> "unresolved"
Lines 161-162. "transportation" --> "transport"
Line 180. "leaves" --> "means"
Line 181. "defines" --> "defined"
Line 187. Comma after "0".
Line 187. "interpret" --> "interpreted"
Line 196. "privde" --> "provide"
Line 200. Remove "based". Rewrite to "This section describes a range of observational
products that are needed ..."
Line 203. Remove "based"
Line 216. "observational estimates" or "observation-based estimates"
Line 218. Would it be clearer to say "As the diffusivities obtained by Groeskamp et al.
(2020 are static, they are ...."?
Line 220. "change at the mixed layer depth is applied" --> "change is applied at the
mixed layer depth" (or maybe "at the base of the mixed layer").
Line 223. Change to "even though they are known to be spatially inhomogeneous" or
"even though they are known to vary spatially"
Lines 225-226. Inconsistent spelling of "parameterization". What is the journal style?
Line 230. Maybe "while maintaining the mixed-layer depth as the separation ...."
Line 244. "interpret" --> "interpreted"
Line 252. "data is" --> "data are"
Line 254. "This data was" --> "These data were"
Line 365. "emphasis" --> "emphases"
Line 366. "of where" --> "in where"
Line 366. "stating" --> "demonstrating"
Line 367. "These results are comparable to, albeit a bit smaller, that found by Griffies
and Greatbatch (2012)" \( \rightarrow$ "These results are comparable to results found by Griffies and
Greatbatch (2012), albeit a bit smaller"
Lines 372-373 and Line 376. "(section 4.4)". These references appear in section 4.4, so
are presumably intended for a different section.
Line 374. "it will be cooler there" --> "sub-surface temperatures will be cooler,"
Line 374. "net smaller" --> "smaller net"
Line 381. "an change" --> "can change"
Line 384. "expansions" --> "expansion"
Line 385. "great ocean" --> "large ocean"?
Line 386. "of an order" --> "that are an order"
Line 400. Line 446. "don't" --> "do not"
Line 404. "between in" --> "in"
```

Line 407, "of same order" --> "of the same order"
Line 414. "their" --> "the"; "is mostly" --> "are mostly"

Line 415. "by means of the different diffusivity parameterisations used" --> "by the different diffusivity parameterisations"

Line 394, Line 418, Line 420, Line 448, Line 484. "extend" --> "extent"

Line 418. "greater ocean basins" --> "major ocean basins" maybe

Line 418. Missing period at end of sentence.

Line 418. "The mesoscale diffusivity used, are" The intent is unclear, but I think this should say, "Mesoscale diffusivities are"

Line 421. "Location comparable to other studies considering cabbeling in the ocean". No verb in sentence. What is the intended meaning?

Line 427. "completeness 5" --> "completeness in Fig. 5" (presumably)

Line 431-432. "Of the remaining two terms, it is only the "production term" Pstir that has a significant impact on GMSL rise"  $\diamond$  "Of the remaining two terms, only the "production term" Pstir has a significant impact on GMSL rise"

Figure 5 caption. "Thermobaricity" --> "thermobaricity"; "the different the" --> "the different"

Line 449. "different method" --> "different methods"

Line 449. "difference" --> "the difference"v

Line 453. "due to" --> "from" to maintain consistent structure within the sentence

Line 457. "positive and negative due to the divergence operator exist" --> "positive and negative change exist due to the divergence operator"

Line 513. "Parameterization" --> "Parameterizations"

Line 543. "can't" --> "cannot"

Line 544. "distributed" --> "distribution"

Line 548. "has has" --> "has"

Line 553. No verb in the sentence starting "Albeit". Instead, "Albeit" should be a connector to the previous sentence, using a comma: "other, albeit over...."

Line 558. "impact on" --> "impact"

Line 576. "are due" --> "due"

Line 604-605. Sentence beginning "Accounting for all" seems to be missing a verb.

Line 627. "assure" --> "assures"

Line 631. "With" should probably be a continuation of the previous sentence (i.e. ", with")

Line 644. "expression" --> "expressions"

Line 695. "In addition use" --> "In addition, we use"

Line 733. "the the" --> "the"

Line 756. "definition" --> "definitions"

---

## Author Comment (AC2)

**Observational-based quantification of physical processes that impact the evolution of global mean sea level**

Sjoerd Groeskamp

**Response to Reviewer #2**

This manuscript attempts to provide a bottom-up observational-based quantification of physical processes contributing to changes in sea level. The study presents a comprehensive set of theoretical equations to describe the underlying processes, which contrasts with the traditional "top-down" budget closure methods. As the author highlighted, the presented work does not aim to close the sea level budget but provides a framework to gain insight into the magnitude, uncertainty, and comparative importance of physical mechanisms influencing sea level rise. While the topic is highly relevant and the methodological approach has merit, several major issues limit its suitability for publication in the current form. I therefor recommend the paper to undergo major revisions.

I want to thank the reviewer for the positive comments that this can become a publication after the suggested revisions. The suggested revision below is numerous and have helped to improve the paper. I want to thank the reviewer for this incredible effort to produce such an exceptional long list of suggestions to improve the paper. It has helped a lot.

**Major Comments:**

- 1.1 The terminology used is inconsistent with the scope of the study. The manuscript repeatedly refers to "Global Mean Sea Level (GMSL)" yet often presents analyses and results more relevant to local or relative sea-level changes. By definition, GMSL implies a globally integrated measure, but the often "local" estimations throughout the paper are not a "global mean."
- 1.2 Also, major sources of the barystatic sea level changes, which have accounted for approximately two-thirds of the total increase in GMSL, are ignored or not well represented in the study.

Thanks, for these comments. With regards to the first comment (1.1). I went through the document and corrected the terminology where I think this was needed. I emphasized local sea level evolution when considering the spatial maps and emphasized GMSL rise or specifically mentioned "after global integration" when discussing the total numbers. In the abstract (L2), the introduction (L44), and section 2 (L91) it is now also specifically mentioned that we will discuss both spatial varying maps of sea level evolution, as well as their global integrals. With regards to the second comment (1.2), I refer to the answer below the next comment.

(2) There is an uneven level of detail provided for various processes, with some described at length and others only briefly referenced. The study is very detailed in ocean processes, but major sources such as land ice melting and terrestrial water storage changes are inadequately represented or completely ignored. Like point (1), this needs to be addressed by clarifying the scope of the study (e.g., be clear that only steric sea level changes are considered) or by widening the scope of data analysis (e.g., include additional data sources to better estimate barystatic sea level component) to allow for a robust and "real-world" GMSL budget.

The reviewer makes the point that barystatic contribution to sea level rise, are not properly accounted for. The barystatic contribution makes up about 66% of GMSL budget over the last century and currently cause about 2 mm/year of GMSL rise. So, it is an essential term for sea level rise budgets. Therefore, it is a good point that the scope of the study can be clarified with

regards to this point. This is done and details are given below. However, this does not ask for more thorough analyses of the mass fluxes themselves, as I will explain below and in the paper.

- 1 Two mass flux products are used. OA and CORE. GMSL rise vary about 40 mm/year between these two products. That is about 10 times as much as current sea level rise, and 80 times as much as the barystatic component. In other words, the uncertainties in estimated GMSL rise due to these products, swamp the imperfect inclusion (or missing) barystatic component. This difference is explained for -65mm/year due to evaporation, 7 mm/year due to precipitation and 19 mm/year due to river runoff. The latter term also includes differences in including the barystatic component due to CORE and OA. Because the overarching problems with these mass fluxes are so large, I do not see reason to further detail the barystatic component into separate contributions from Antarctic or Greenland ice melt or TWS. Instead, the message is about the fact that these uncertainties are so large and that we need to better constraint them. I believe this paper provides a first substantial calculation of this kind from observations.
- 2- The fact that other terms seem more detailed, basically just rolls out of the derivations and helps to connect to existing literature. For example, the impact of mixing has many terms due to the rewrite into the divergence/redistribution term and production term. However, many of these terms can be neglected and the main point is about the production terms. This is clearly indicated but provided for completeness. To connect these terms to existing literature, the difference between horizontal, vertical and neutral mixing also needs to be detailed. Together, this leads to a lot of detail. This detail is thus more from a "derivation" perspective and not so much from the products used. Ass with the mass flux. I do not dive into the details of the different mixing products. Like mass fluxes, they are given "as is". Which, for the scope of this study, should be enough.

The purpose of this paper is thus not to consider the details of all the products used (also think of details of mixing products or heat flux products), but instead take them as they are, and gauge the uncertainties and magnitudes of the terms. I think the title also clearly indicated that the physical oceanographic processes are the focus. However, to avoid confusion, focus expectation and don't provide false promises, I have rewritten parts of the paper to further emphasize the point the reviewer makes. Here is a list of the most important changes regarding this issue:

- The term barystatic is now introduced in the second paragraph of the introduction (L28-L30)
- L5 (abstract), L42-L48 (introduction) states the budgets can't be closed, that the goal is to measure the uncertainty / range in magnitude and the goal is to compare the uncertainties between products and processes.
- L287, section 3.5 (data) it is mentioned that barystatic contribution is not well represented and that this is not studied in detail.
- L329, section 4.1 (results) now discusses the fact that the barystatic signal is swamped by other uncertainties.
- L474, section 5 (discussion) now discusses and re-iterates the fact that the barystatic signal is swamped by other uncertainties.
- L531, section 6 (conclusions) now mentions the barystatic component, but not the issue about its representation.
- L548, section 6 (conclusions) this part clearly states that mass flux products have the uncertainties and make sure we can't close the budgets and that improving these products is beyond the scope of this study.
- L554, section 6 (conclusions) the summary clearly states that mass flux products have uncertainties and cause that we can't close the budgets.

I hope this puts the issue at rest. By contrasting the issue of the missing barystatic component against other errors. Also indicating how far we are from being able to close the budget this way. And hopefully the reviewer agrees with me that this assessment is useful for addressing this point. Perhaps it could help to motivate for improved observations and theories to constraint the mass fluxes that feed our ocean models. I mention this in the conclusions.

(3) There are many instances of unclear, awkward, or grammatically incorrect writing that hinders readability and communication of the scientific results. Besides the issues pointed out in the line-by-line comments below, I recommend revising the language in general to be clear, concise, and well structured as possible.

I thank for the reviewer's unbelievable effort to point at all these writing issues laid out below. I have implemented all suggested improvements. This includes:

- "Parameterisation" is now written with a "z" everywhere.
- All "I" and "we" is removed.
- I removed most cases using "for example".
- I removed "significant" in various places other than noted by the reviewer.
- I reduced the use of "xx" to only the first time a term is introduced.
- I have gone through the manuscript carefully to improve writing.
- Shortened the abstract.

All in all, I think the paper has substantially benefitted from these efforts all together. I hope the writing is no longer in the way of the messaging.

**Line-by-Line Comments:**

As a consequence of the below points made by the reviewer, the abstract has been substantially rewritten and shortened.

- Lines 1-2: Clarify what is being integrated. Current phrasing is ambiguous. Suggest specifying components involved (e.g., ocean heat content, satellite altimetry). Also, please clarify the distinct roles between data sources and methodology. Altimetry measures sea surface height; mass balances (e.g., GRACE) capture contributions from land ice. What models are referred to?
- Line 3: It's unclear what "such methods" is referred to. I assume this is what will be referred to as the top-down approach. It would be best to name this upfront.
- Lines 4-5: process --> processes

  Also, the whole sentence is basically a list of example processes. Such a phrase seems unnecessary for the Abstract. I recommend moving it to the Introduction
- Line 7: Delete "and"
- Line 8: Awkward phrasing. A clearer version would be: "It is neither the intention nor is it possible...". Remove comma (,)
- Line 9: Add comma after "Instead"
- Line 10: Change to "and allow comparisons of the impact of individual processes or parameterizations on GMSL"

Line 18: Missing comma (Since 1850,). Redundant wording in "Human induced anthropogenic warming" --> "Human induced" and "anthropogenic" is the same thing. Choose one. Good point! Done.

Lines 18-19: The 2nd sentence is repetitive. Suggest rephrasing: "Since 1850, human-induced warming has increased global surface temperatures by about 1.1 °C (IPCC, 2021). About 89% of this anthropogenic heat has been absorbed by the ocean..."

**Nice!**

Line 21: Clarify whether the 4% warming is also anthropogenic warming and constitutes in addition to the 89%? Where did the rest 7% go?

I added a sentence.

Line 24: Remove duplicated "mm". Done.

Line 24-25: Masson-Delmotte et al. (2021): Please use consistent citations. Here the Author et al. is used but it is referring to IPCC, 2021. which was cited in the beginning of the paragraph: IPCC: Climate Change 2021: The Physical Science Basis. Contribution of Working Group I to the Sixth Assessment Report of the Intergovernmental Panel on Climate Change, vol. In Press, Cambridge University Press, 2021. In general, I recommend citing chapters of the IPCC report with the corresponding " et al." and not the whole report. Done.

Line 27: "integrating over global temperature and salinity budgets" - This does not measure the GMSL. It estimates the steric sea level, which is only one part of the GMSL.

Thanks. I've changed the sentence to reflect this comment.

Line 28: "using satellite altimetry" - I assume this refers to measuring GMSL, which is the sum or steric and barystatic (ice sheet+glaciers+terrestrial water). This should be clearly phrased. You are using a mix of data sources (altimetry) and methodology (integration over global temperature).

I see the confusion. I think the rewrite should be correct now.

Line 29: Team, 2018 is not the correct citation: This should be Shepherd et al. (2018). Shepherd, A., Ivins, E., Rignot, E., Smith, B., Broeke, M. van den, Velicogna, I., et al. (2018). Mass balance of the Antarctic Ice Sheet from 1992 to 2017. Nature, 558(7709), 219–222.

https://doi.org/10.1038/s41586-018-0179-y

I'm sorry. This is how it was downloaded from the Nature website. I now have a new version with names.

Line 31: The sentence is awkward. Firstly, the shift to first-person narration ("I here apply...") is odd and inconsistent with the otherwise formal tone of the manuscript e.g., "This study estimates..."). Secondly, sentence structure makes it hard to read. I would refrain from using first person and combine this sentence with the next: "Instead of the top-down approach, this study adopts a more 'bottom-up' strategy by estimating GMSL as the sum of contributions from individual physical processes that alter ocean density and, consequently, ocean volume." I agree and have consequently done a broader rewrite of this section of the introduction, including the suggestions above.

Line 40: model -> models. Done.

Line 43: To be consistent with spelling "parameterisations" should be "parameterizations". Done.

Lines 46-48: "Results show..." Repetitive. This was already stated in the previous paragraph. I removed this part and merged it with the paragraph above.

Line 49: The phrase "blowing up" is an interesting choice of words. I don't mind informal terminology, but I fear it is rather confusing than helping understand the underlying concept. Consider rephrasing this to clarify precisely what you mean. Done.

Line 71: amongs -> amongst. Done.

Line 77 (Equation 1): Clarity that  $\rho(\epsilon)$  refers to the local density at the sea surface. It would probably better to call it  $\rho(z = \epsilon)$  or  $\rho(\epsilon)$ . Otherwise, it could be confused with  $\epsilon$  reta, or rho being a function of reta. Done.

Lines 78-79: The sentence ("Here...... changes in density") is not comprehensible. A word seems to be missing here. Done.

Lines 79-80: Rather than listing examples, which appear arbitrary or misclassified, I suggest clearly stating the physical processes or mechanisms the "latter term" represents. Referencing Appendix A is helpful, but the main text should still provide a coherent conceptual definition. Done.

Line 83: "inverse barometric effect" -> "inverse barometer effect". Done.

Line 84: Based on the title, I thought the focus here is on the global mean. Why focus on "the spatial structure"? The terminology describing the sea level changes should be precisely defined. If relative sea level changes are presented, then this should be reflected in the title and the terminology used throughout the paper.

I believe the spatial structure is valuable to show and helps understanding and interpreting the global mean term. I therefore find it valuable to leave these results in the paper. However, I clarified the scope of the paper and improved terminology (see response to major comment 1.1).

Line 85: Unclear what "its" refers here. Done.

Lines 90-95: This paragraph has a confusing structure and is hard to follow. I have improved this paragraph.

Lines 93-94: Change to: "Derivations rely heavily on Griffies and Greatbatch (2012) and Groeskamp et al. (2019b)." Done.

Lines 94-95: Unnecessary statement. Just refer to the appendix when discussing the relevant derivations. No need to generally announce it or justify it. Done.

Line 95: Last sentence is not clear. What is meant with "main points" vs "new points"? Consider removing these general statements and instead clearly describe the points you want the reader to take away. Done.

Subsection 2.1 title: change to flux -> fluxes. Done.

Lines 96-99: Ensure readability when including equations in the text. Please see the suggested revision:

At the ocean boundaries, the ocean mass flux  $Q_{\text{mass}}$  (in kg s-1 m-2) is defined as positive into the ocean and given by:

 $Q_{\text{ext}} = P - E + R + I + A_e, (3)$

where P is precipitation (positive into the ocean), E is evaporation (positive out of the ocean), R is runoff from rivers(positive into the ocean), I is runoff from land ice melt (positive into the ocean), and A\_e is the aeolian deposition of salt (positive into the ocean). Nice rewrite. Done.

Line 101 (Equation 4): I am confused by the term "I" in the equation. Sea ice should be irrelevant for sea level change. The missing and important piece is the ice sheets. Is this what should be represented by "ice melt" or assumed to be part of runoff?

It should be land ice melt. However, I agree this can be somewhat arbitrary as some of that might also end up in rivers. It depends on the mass flux product used. I like your rewrite using land ice melt specifically (as it as intended) and will keep it that way. For further clarification I refer to major point 2.

Line 104: Add period at the end of sentence. Done.

Equation (5): Clarify the integral limits in the text. It is integrating from the ocean bottom (at depth H) to the ocean surface (at height \eta).

I have instead added this below Equation 1, where the limits were also not specified.

Line 108: What part is shown in Appendix A and what part is shown in Appendix B? Done.

Lines 108-109: Sentence is convoluted and difficult to follow. "contains the impact" is unclear and vague. "therewith density" is awkward phrasing. I suggest rewrite to "The term  $F^{\text{no}}(\text{mass}) (kg m^{-3} s^{-1})$  represents the changes in local ocean density due to ocean mass fluxes (Q{\text{mass}}), primarily through alterations of local salinity." Thank you for the suggested rewrite. Done.

Line 109 "Mainly by..." This is an incomplete sentence. I suggest to merge it with the preceding sentence. Done

Line 110: What are sources of salinity? Do you mean salt and freshwater? I added some hypothetical sources, with the comment that such sources are generally considered negligible.

Line 111: Add period after sentence. Done.

Line 112-113: Change to "The impact of shortwave radiation on ocean density is represented by the term  $F^{\left( \right)}_{\left( x\right)} (kg m^{-3} s^{-1})."$  - Does that mean that shortwave fluxes are not included in the surface heat fluxes that are considered in  $F^{\left( x\right)}_{\left( x\right)} (text{surface})$  or only the non-penetrative part of SWR? Please clarify. Done.

Lines 113-114: Awkward phrasing ("associated with for example") and there should be an explanation why it can be "ignored". Clarify and elaborate: "Surface salt fluxes (e.g., due to sea ice melt or sea spray) are not considered in this analysis, because..." It's part of the rewrite of this paragraph.

Line 115: change to: "which has a relatively minor impact on ocean density (de Lavergne et al., 2015)." Done.

Line 118: are -> is. Done.

Line 119: I don't think the sentence is clear. Better to spell it out: Eddies on the order of meters. Done.

Line 119: Here, and throughout the manuscript, there is an overuse of the phrase "for example," which weakens the clarity and precision of the text. Instead, I suggest directly focusing on the most common processes without relying excessively on illustrative phrasing. Suggested rephrasing: "In this study, small-scale mixing refers to processes at scales on the order of meters, most commonly associated with breaking internal waves and boundary-layer turbulence (MacKinnon et al., 2013; Large et al., 1994). This mixing is represented by a vertical turbulent diffusivity \$D\$, acting on vertical tracer gradients (McDougall et al., 2014)." I have changed this occasion and went through the manuscript to reduce the use of "for example".

Line 129: "directional and scale variations" - I am not sure what this means. Consider rephrasing. Done.

Lines 129-130: Repetitive use of the term "tensor". Change to: "represented by a symmetric and positive-definite diffusivity tensor". Done.

Line 136: Incomplete sentence. Should be combined with the previous sentence+equation to improve readability of the text: ", where the following definitions (see Appendix C) apply:" Done.

Line 146: Remove "is" between "impact" and "on density". Done.

Line 148: \Theta and S\_A need to be defined. Earlier text just refers to temperature and salinity. Usage of symbols should be consistent throughout the text. Done under equation 8, where it is first used.

Line 149: "Here the same naming" - The same naming as what? Done.

Line 150: "three mixing direction" - What are those three mixing directions? Done.

Line 157: I don't see any eddy velocity in Equation 1. That is because this is explained just above Eq. 9. I have added some clarification.

Line 159: GMSL (Global Mean Sea Level) explicitly implies a global integral. A "local" estimation, by definition, is not a "global mean." I think I clarified this.

Line 160: "will be zero when globally integrated" - That means that dynamic changes have no impact on the GMSL budget. So why include them in this analysis if the focus is on GMSL? restructuring into non-divergent components is also a large part of the manuscript. I find it valuable to show the results to confirm this, but also for future reference and completeness.

Lines 161-162: The phrase "non-resolved transportation" is typically used in the context of numerical ocean modeling, where the model grid is too coarse to explicitly capture smaller-scale processes. How is this relevant in the context of this study, which is based on observations?

Indeed, unresolved could be considered model only. I changed it to "unrepresented" instead of unresolved. This makes it slightly more general. Using a 1-degree grid observational based product, means that the geostrophic velocity does not capture these eddy velocities. As with a model.

Lines 180-181: The sentence is awkwardly phrased and grammatically incorrect. I have rewritten this part, including surrounding sentences.

Line 186: ... of \*an\* incorrect neutral... Done.

Line 187: interpret -> interpreted. Done.

Line 191: overestimate -> overestimation. Done.

Line 191: "also lead to more reduction in GMSL" - Why "aslo". As compared to what other process? Changed.

Line 196: privde -> provide. Done.

Lines 197-198: Please elaborate what "discuss" mean? Section 4 is the result section. So it should say that the processes are estimated and presented in section 4 using data from section 3. Done.

Line 200: It would help to explicitly list the observational variables (e.g., temperature, salinity, pressure, velocity fields, fluxes) in a table along with the datasets used. This would give readers a quick and clear overview of the observational products utilized for calculating the GMSL budget terms defined in Section 2.

That is a good point. A table is added to section 3.

Line 203: "For the observational based climatology," - Isn't all of the Data observation based? This sounds like both observations and model based climatologies are used. What are the variables? Done.

Line 204: 'in situ temperature, t' - use commonly used symbols that are consistently used throughout the manuscript.

Here t is the symbol that we use for in-situ temperature in literature (see also TEOS-10 manual). Theta is the symbol for Conservative Temperature. The point is that in-situ temperature is given and needs to be recalculated to \Theta, as described in the text. So in this case the used terminology and symbols are correct.

Line 203-205: This sentence can be made less convoluted by focusing on the relevant variables used. No need to refer to "other tracers" or speak of annual / seasonal if monthly is used. Done.

Line 208: Instead of "IOC", use auhor name et al. to be consistent with citations on other literature. At least it should be IOC, SCOR and IAPSO, 2010 or better citation would be Roquet et al., (2015)

I understand the comment, but in this case, I will not change it. The TEOS-10 manual has no official "authors", and the IOC et al is how it is mostly cited. This is how the journals also treat the citation. Roquet is not the author for sure (although he did contribute at a later stage).

Line 209: Specify what variables are used. What are the inputs? Done.

Line 210: How is mixed layer calculated? What criteria is used? Done.

Line 210: Define "Static stability" It is defined as (N^2>0), but I added "stably stratified water column" for further clarification.

Lines 211-210: "TEOS- 10 software" is a package of algorithm. Specify what algorithm (subroutine) is used along with the relevant publication. I did where addition publications are justified, such as for obtaining a stably stratified water column, or for the mixed layer depth routine. In all other cases the citation of the published software package Barker and McDougall 2011, covers the information needed to reproduce the results.

Line 220: Add comma between "depth" and "mixing". Done.

Line 221: "mesoscale mixing" - Repetitive phrasing. Change to mesoscale eddies? I don't quite see what is meant here.

Line 221-222: "The same product" - Clarify what data product you are referring to. Done.

Line 226: parameterisation -> parameterization. Done.

Line 227: Avoid overuse of "for example" and use specific terminology to describe the relevant methodologies. Done.

Line 228" "significant" implies a statistical test which is not relevant here. Done.

Lines 226-229: Elaborate on the role of boundary layer mixing. I suggest to divide the sentence and provide quantification of the impacts based on Griffies and Greatbatch (2012). Done.

Line 231: (subsection title): Slopes -> slopes. Done.

Line 235: specify what tracer. Done.

Line 241: Avoid double parenthesis. Done.

Line 242: Elaborate and add references that actually show that "most models still use the "local" method". Done.

Line 244: Howver -> However; interpret -> interpreted. Done.

Line 248: Besides naming the routine ('gsw\_geo\_strf\_dyn\_height'), the underlying concept should be introduced along with relevant citations. What is the difference between the GSW Software (referred here) and the TEOS-10 Software (referred earlier)? Good point. They are the same thing, I just used inconsistent naming. I now made it consistent.

Lines 253-254: All of this details should be moved to the Open research / Data access section at the end. Done.

Line 257: Don't think it is necessary to have with this single sentence. I think I removed the right sentence and rewrote the surrounding.

Line 261: Add period after "1983-2006". Done.

Line 265-266: Justification is needed. Provide sources that justify that "spreading the other 50% over the surrounding ocean." is viable. There are no sources, this is a personal choice. For SLR calculations the effects will only be in the fact that surrounding grid points may have a tiny bit different beta-value, such that the impact on density could be different. I could also have put everything in the one bin where the river mouth ends in the ocean. Honestly, the difference can only be small.

Line 267: The statement needs a proper reference instead of pointing to the dataset name CORE2. Done.

Line 271: Should the units be W/m^2? Thanks! Done.

Line 272: Define P, or keep using precipitation. Done.

Lines 272-273: Why use upper case for "Latent Heat, Sensible Heat, Longwave" but not for "shortwave radiation, evaporation and precipitation". No clue. Changed it!

Line 274: "Runoff is based on (Dai, 2016)," This has already been stated. No, that was for the OA product, not for the CORE product.

Line 277: Change to: (see Fig. 2 in Valdivieso et al., 2017). Done.

Line 278: What are the overlapping datasets? Rewritten.

Lines 283-290: I don't think it is necessary to provide this material as a separate section. Merge with Section 3.5. I made it a subsub section, as I find it important to keep highlighting that these products are different as they are artificially balanced.

Line 294-295: Incorrect sentence structure makes this incomprehensible. Done.

Lines 302-303: Seems kind of overkill to provide so much detail about the other parameterizations in the main text if they are not used in the main analysis. A lot of this is now in the appendix.

Line 303: paramterization -> parameterization. Done.

Lines 304-307: This is hard to follow. Try breaking up into shorter sentence. Done.

Line 307: "This provides..." - What is "This" refer to here? Done.

Line 309-310: I assume the different dataste have been subsetted to a common time span. At the beginning of this section (or in the previous section), clearly state over what time period the analysis is done. All data sets are given or calculated into a "standard year" for reasonably corresponding periods.

Line 314: Sign convention seems counterintuitive and inconsistent with Figure 1. It should follow the convention described in Section 2.1 Thanks for picking this up.

Figure 1: You used mm/year in the main text. Why do you use m/s here? It should be the same units throughout the paper. Done.

Figure 1 (caption) "Runoff from ice melt" - Do you mean land ice? Please specify. Ice sheets are one major contributor of sea level rise. It strikes me as problematic if one of the major source of mass to the ocean is being ignored. See the response to major comment 2.

Figure 1 (caption): Specifiy the data sources for each panel. Done.

Table 1: "Core". In the main text, you refer to CORE. Please be consistent with the naming. Done.

Table 1 (caption): "GMSL rise in mm year-1" - Over what time period?

Results are obtained from a "standard year hydrography". To avoid confusion and clarify this, this is now mentioned at several points for several datasets. Beginning in the introduction, immediately stating how the results should be interpret, giving:

- L43, Introduction
- L215, section 3.1.
- L282, section 3.5.
- L296, section 3.5.1.
- L312, section 3.6
- L317, section 4.

A time evolving analyses could be made in the future. For example, using reanalysis products. However, this is beyond the scope of this study.

Table 1 (caption): Rephrase for better readability: "Mass and freshwater fluxes are due to evaporation, precipitation, and runoff, while heat fluxes include radiative fluxes (longwave and shortwave) and turbulent fluxes of latent heat and sensible heat." Thanks. Done.

Line 318: 10 times -> ten times. Done.

Lines 322-323: Antarctica runoff is not a negligible source of freshwater that can not be left out when using OA flux. It should be quantified what is missing in OA flux compared to CORE. The analysis should provide an estimate of the missing Antarctica runoff, otherwise the CORE vs OA comparison is an apples vs. oranges comparison.

I refer to my response to major comment 2.

Line 323-324: Any sea level analysis that does not represent Greenland and Antarctica sources (whether present day or future) is insufficient and strikes me as an unnecessary excercise. I refer to my response to major comment 2.

Lines 331-345: If the study is concerned with only the GMLS, this section seems unnecessary. Specifically, as explained by Gregory et al. (2019) in their Appendix 2, global halosteric sea-level change is negligible and can generally be ignored in GMSL assessments.

As Griffies and Greatbatch, we find that the impact is of the order of 0.3 mm/year. This is not negligible and worth putting in the paper. This simply rolls out of the equations + application.

Line 338: Add period after (Fig. 2). Done.

Figure 2 (caption): "Runoff from ice melt I" - same comment as in Figure 1. Done.

Line 344: Avoid word "significant" unless statistical test is presented that show significance. Done.

Line 348: And what about SWR that is absorbed at the surface? Also, cite the subsection in section 2 (it is not discussed in section 2.2). I have rewritten this part. SWR is in fact mentioned in section 2.2. But I now also further reference section 3.6 for completeness.

Line 350: "highest" relative to what? Precipitation and evaporation are >1000. Improved. Thanks.

Lines 353-356: "while imbalances a 100 times" - Instead, state the estimated imbalances that cited studies found ( $\sim$ 30 W m $\sim$ 2). Done.

Line 361: a factor \*of\* 10. Done.

Line 365: Add the estimated GMSL rise in parentheses. Done.

Line 366: Clarify what "these results" refer to. Done.

Line 367: "albeit a bit smaller, that found " - Change to "albeit a bit smaller than what was found". Done.

Line 373: no comma between "surface" and "is". Done.

Line 375-376: What component (row in Table 1) are you referring to here? I do not see specific row for SWR. It's there. Now specifically added row 11.

Line 377: Remove "level". Done.

Lines 379-380: Elaborate why a deeper penetration of SWR must give lower estimates of thermal expansion. The first few sentences of the paragraph were dedicated to explaining this.

Lines 380-382: This sentence is not comprehensible in the current form. Done.

Lines 383-389: Given the small magnitude and impact, it strikes me as odd to give impacts to geothermal heating a separate section but completely ignore the effect of terrestrial water storage, glaciers and ice sheets. I understand, but in this case, I would like to keep it this way. IT also showcases the method and provides very similar results as Griffies and Greatbatch 2012, which I find satisfying.

Line 385: The manuscript currently cites Figure 5h before Figure 4 has been introduced, and the main discussion of Figure 5 appears two sections later. This disrupts the logical narrative and readability. Please ensure all figures are cited and discussed sequentially, or adjust the figure order to align clearly with the text. Fixed.

Figure 5 contains multiple small panels with different scales that are difficult to compare. Includes geothermal heating term which is discussed before Figure 4. Fixed.

Figure 5 (caption): delete duplicated "the" in "the the diffusion-density interaction". Done.

Figure 5 (caption): "Note the different the color bars." is not a correct sentence. Done.

Lines 398-399: This belongs to either the Introduction or Discussion section. It is now moved to the discussion. Line 493.

Line 401: Ending of the sentence is unclear. Done.

Line 402: factor \*of\* 2. Done.

Line 404: factor of 5 between what? Done.

Lines 404-406: This study needs to provide a quantification of the error. What is the likely range in GMLS rise due to the uncertainty in mixing schemes?

The likely range seems impossible to obtain, there are so many unknwon errors involved in all the included products. Instead, the way to do this, is to use different products for diffusivities. And in case of neutral gradients, the method related to how these calculations are done. One could even use different hydrography's (which is not done in this study, and beyond the scope here). Then one gauges the range of results, coming from these different products. That provides an idea of the magnitude and uncertainty related to these choices and products. I tried to further clarify this in the first lines of section 4.6. I also changed the word uncertainty to difference.

Line 407: -3 and -11 mm per year? Done.

Line 408: parameterisations -> parameterizations; of \*the\* same order. Done.

Line 415: parameterisations -> parameterizations. Done.

Line 418: Missing period (.) after "(Groeskamp et al., 2019a)". Done.

Line 418: No comma between "used" and "are"

Line 421-422: Incomplete sentence. It should specify (describe) the similarities with the cited studies. Done.

Line 424: Period missing. Done.

Line 427: "are shown for completeness 5)" -> in Figue 5? Done.

Line 438: "explained in section 3" - specify subsection. Done.

Line 439: are -> is. Done.

Line 446: Provide statistical test or rephrase. Done.

Line 446: discus-> discussed. Done.

Line 450: No comma between "method" and "makes". Done.

Line 456: Consider rephrasing to clarify what Figure 4 in Griffies and Greatbatch (2012) actually show. Done.

Line 467: 10 times -> ten times. Done.

Line 468: sum of the parts -> sum of contributing processes? Done.

Line 469: add i.e., before "bottom-up approach". Done.

Line 480: Please show statistical test or rephrase. Done.

Lines 481-482: "first-order" seems to be overstating the impact. It is only in the context of the hypothetical case of artificially balanced flux. In reality this is never the case unless averaging over the long term preindustrial period. Indeed, the real world the flux has an imbalance of about 0.3 W/m2 (see discussion in section 4.3). CORE and OA have 10- and 100-times larger imbalances. Hence, the balanced products come closer to the real ocean heatflux that the actual products. At least for the net input. The resulting impact of nonlinear thermal expansion is then of the same order as observed GMSL rise. That is what I call first order, as also summarized in the conclusion section (L545).

Line 513: Explain why heating colder waters must lead to less sea level rise. Done.

Line 516: I don't see this quantification of "10 times per difference" clearly laid out in the result section. How is the "difference between climate models" estimated here? Done.

Lines 517-520: This last paragraph is not really Discussion material. It is a technical detail that should be moved to the Method section. Done.

Line 517: "last term, Eq. A1" - You mean, it is the last term in Eq. A1? Done.

Line 519: 1000-10.000 -> 1000-10,000. Done.

Line 522: The statement is repeated from Introduction, but the associated citations are different. Ludwigsen should also be cited in the Intro. And the papers cited in the Intro should be cited here. Done.

Lines 525-527: The English in this sentence is not correct. I assume it should be: "The focus is particularly on the impact of diffusion, stirring, neutral physics, shortwave radiation, and boundary fluxes, all of which alter oceanic density and thus affect GMSL." Thanks!

Lines 527-528: Why are direct mass fluxes not a focus? The barystatic components in GMLS are a major contributor of present-day and future sea level rise. There needs to be a change in termiology: The study should not gall it GMSL. Instead the focus then should be on steric sea level, or global mean thermosteric sea level (GMTSL).

Line 530: You can't call it "global sea level rise" if contribution of land ice are not well defined in the observational analysis.

Line 530: Remove duplicated "and". Done.

Line 544: distributed -> distribution. Done.

Line 548: parameterisations -> parameterizations. Done.

References:

Gregory, J. M., Griffies, S. M., Hughes, C. W., Lowe, J. A., Church, J. A., Fukimori, I., et al. (2019). Concepts and Terminology for Sea Level: Mean, Variability and Change, Both Local and Global. Surveys in Geophysics, 40(6), 1251–1289. https://doi.org/10.1007/s10712-019-09525-z Roquet, F., Madec, G., McDougall, T. J., & Barker, P. M. (2015). Accurate polynomial expressions for the density and specific volume of seawater using the TEOS-10 standard. Ocean Modelling, 90, 29–43. https://doi.org/10.1016/j.ocemod.2015.04.002